# STORIES: learning cell fate landscapes from spatial transcriptomics using optimal transport

Geert-Jan Huizing ®[1,3], Jules Samaran ®[1,3], Daniele Capocefalo[1], Anna Audit ®[1], Gabriel Peyré ®[2] & Laura Cantini ®[1]✉

In dynamic biological processes such as development, spatial transcriptomics is revolutionizing the study of the mechanisms underlying spatial organization within tissues. Inferring cell fate trajectories from spatial transcriptomics profiled at several time points has thus emerged as a critical goal, requiring novel computational methods. Wasserstein gradient flow learning is a promising framework for analyzing sequencing data across time, built around a neural network representing the differentiation potential. However, existing gradient flow learning methods face challenges in analyzing spatially resolved transcriptomic data. Here, we propose STORIES, a method that uses an extension of Optimal Transport to learn a spatially informed potential. We benchmark our approach using three large Stereo-seq spatiotemporal atlases and demonstrate superior spatial coherence compared to existing approaches. Finally, we provide an in-depth analysis of axolotl neural regeneration and mouse gliogenesis, recovering gene trends for known markers such as *Nptx1* in neuron regeneration and *Aldh1l1* in gliogenesis and additional putative drivers.

Spatial transcriptomics technologies are revolutionizing the study of how cells organize within tissues[1]. Techniques based on high-throughput sequencing have enabled the unbiased discovery of gene expression patterns within their spatial context. For instance, recent studies have revealed previously unknown spatial organization at the tumor–microenvironment interface in melanoma and Alzheimer's disease amyloid plaque microenvironment[2,3]. Widely used spatially resolved sequencing techniques (for example 10x Visium) often measure spots larger than the typical cell size. However, recent technological developments based on barcoded arrays like Stereo-seq and HDST have reached single-cell resolution, effectively bridging functional and structural characterizations of the cell[4,5]. Recent works have leveraged Stereo-seq to produce large spatiotemporal atlases of various biological processes by profiling a system with spatial transcriptomics at several points in time[4,6,7]. These datasets are ideal for studying cellular dynamics within the tissue during processes such as development and the onset of complex diseases, where cells undergo coordinated transcriptomic changes and spatial reorganization.

Inferring the dynamics of biological processes from single-cell sequencing data requires tailored computational approaches known as trajectory inference methods[8]. Monocle initiated the field of trajectory inference by ordering cells along a pseudotime axis based on their transcriptomic similarities and analyzing gene expression trends along pseudotime[9]. While pseudotime represents the progression along a differentiation process, pseudotime-based methods do not provide a model for the underlying transcriptomic changes and thus cannot predict a cell's future transcriptomic state[10]. RNA velocity has thus been proposed to predict changes in gene expression based on splicing dynamics[11]. However, velocity-based methods rely on simplified kinetic

[1]Institut Pasteur, Université Paris Cité, CNRS UMR 3738, Machine Learning for Integrative Genomics Group, Paris, France. [2]CNRS and DMA de l'Ecole Normale Supérieure, CNRS, Ecole Normale Supérieure, Université PSL, Paris, France. [3]These authors contributed equally: Geert-Jan Huizing, Jules Samaran. ✉e-mail: laura.cantini@pasteur.fr

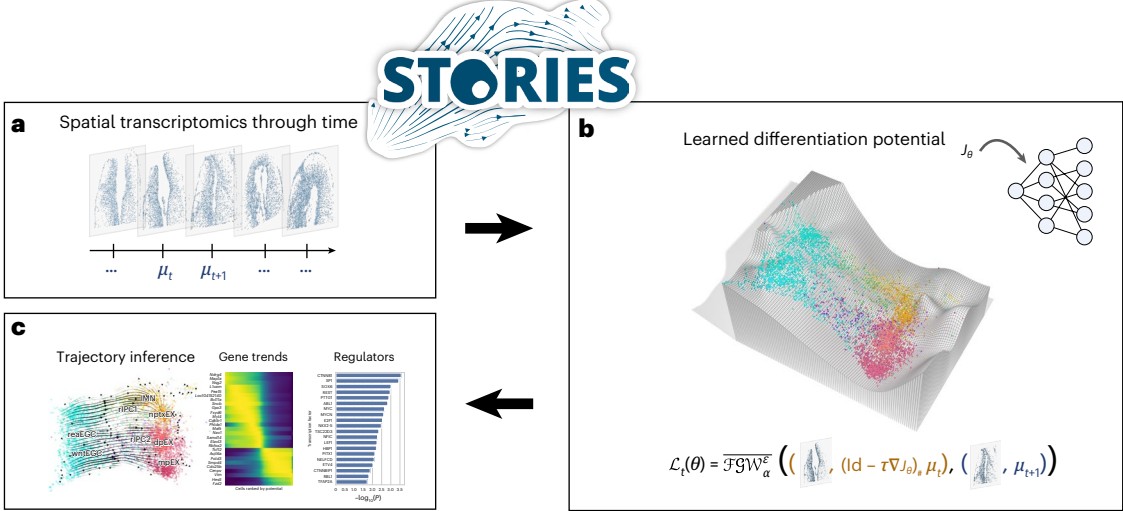

**Fig. 1 | Overview of STORIES. a**, STORIES takes as an input spatial transcriptomics through time. **b**, STORIES learns the parameters $\theta$ of a neural network $J_\theta$ representing the differentiation potential of a cell based on its transcriptomic profile. The objective function is based on FGW, which leverages both the transcriptomic profile and the spatial coordinates. **c**, The gradient of the function $J_\theta$ delivers a velocity that can be used to perform trajectory inference. The potential itself is a natural alternative to pseudotime and allows the study of gene trends along differentiation. Finally, STORIES can highlight possible transcription factors regulating differentiation.

models that can misinterpret cell dynamics, for instance in the case of transient boosts in transcription[12].

Multiple methods based on Optimal Transport (OT) have been developed for cases when several time points are available along differentiation. Waddington OT infers trajectories by computing probabilistic cell–cell transitions between adjacent time points[13]. However, it delivers neither a notion of pseudotime nor a notion of velocity. Another class of OT-based methods proposes a continuous model of population dynamics by training neural networks representing a generalized notion of velocity[14]. However, these methods do not order cells along a pseudotime axis. A promising OT-based framework for trajectory inference consists of learning a potential function governing a causal model of differentiation[15–17]. Framing cellular differentiation as the minimization of a potential function is rooted in systems biology and formalizes Waddington's idea of epigenetic landscape[18,19]. Furthermore, the potential function is a natural alternative to pseudotime, and its gradient yields a rigorous notion of velocity.

The recent development of spatial transcriptomics technologies made it possible to study how space impacts cell trajectories. The addition of spatial information to classical trajectory inference methods is extremely difficult as obtaining a representation for spatial coordinates that is coherent and comparable across all time points is challenging. Indeed, spatial coordinates cannot be used as input by some methods the same way gene expression is used because of possible rotations and translations occurring on measured slices. Furthermore, when studying trajectories during development, living organisms can undergo important morphological transformations. A few OT-based approaches inferring trajectories from spatial transcriptomics through time have recently been developed[20–22]. For instance, stVCR learns a spatial velocity along with a gene expression velocity after rigidly aligning the slices. However, this alignment can only partially capture the morphological modifications described above. Additionally, both velocity terms in stVCR depend on the time, this can allow the model to learn internally a better spatial representation for each time point. This strategy might impair stVCR's generalization performance on unseen slices of later time points. Moreover, SpaTrack learns velocities based on linear OT taking into account both space and gene expression. Moscot computes cell–cell transitions between adjacent time points using an extension of OT called Fused Gromov–Wasserstein[23] (FGW). FGW is properly suited to compare slices across time points since its

transport plan is invariant to rotating, translating and rescaling of the spatial coordinates. However, SpaTrack and Moscot only connect cells from adjacent time points, but cannot predict the evolution of cells at unseen future time points.

Here, we propose STORIES, a trajectory inference method capable of learning a causal model of cellular differentiation from spatial transcriptomics through time using FGW. STORIES uses FGW as a machine learning loss to learn a continuous model of differentiation. FGW allows STORIES to implicitly guide the learned potential to depend on the space information without having to use a representation of the space as a direct input. We can therefore learn a differentiation potential shared across all time points and model a general dynamic less prone to overfitting. This potential can then be used to predict the evolution of cells at future time points.

We benchmarked our approach on three large-scale spatiotemporal Stereo-seq atlases, covering mouse development[4], zebrafish development[6] and axolotl regeneration[7] showing superior performances over state-of-the-art methods. Furthermore, we used STORIES for the in-depth analysis of cellular trajectories in axolotl neural regeneration and mouse gliogenesis. In both cases, we show that the spatial environment impacts cell fate decisions. We then recover gene trends for known markers, such as *Nptx1* in Nptx+ excitatory neuron regeneration and *Aldh1l1* in gliogenesis. In addition, STORIES uncovers other possible driver genes and transcriptional regulators of cellular differentiation in these contexts, which may be of interest for further biological investigation.

Finally, we provide STORIES as an open-source and user-friendly Python package[24]. It is based on the Scverse ecosystem, making it easy to interface STORIES with existing tools for single-cell analysis such as Scanpy and CellRank[25–27]. In addition, STORIES benefits from the JAX ecosystem for deep learning and OT computation, enabling the fast handling of large datasets[28,29].

## Results

### STORIES: a spatiotemporal cell trajectory inference method

We developed SpatioTemporal Omics eneRgIES (STORIES), a tool for single-cell trajectory inference using omics data profiled through spatial and temporal dimensions[24]. STORIES allows studying dynamic biological processes in their spatial context by identifying cell fates, gene trends and candidate transcriptional regulators (Fig. 1).

STORIES is based on the OT, a mathematical framework that enables the geometrically meaningful comparison of distributions, using various flavors of the Wasserstein distance[30]. OT also provides a valuable model for population dynamics: the so-called Wasserstein gradient flows were popularized by Jordan, Kinderlehrer and Otto for their connection with the Fokker–Planck equation and were recently used for trajectory inference in single-cell transcriptomics[15–17,31]. However, existing methods for trajectory inference based on Wasserstein gradient flows are not equipped to deal with spatially resolved omics data. STORIES introduces key methodological innovations that allow one to address the specific challenges of including spatial information.

As an input, STORIES takes slices of spatial transcriptomics profiled at several time points. For instance, Fig. 1a displays sections of axolotl brains profiled at different stages during regeneration. STORIES then learns the parameters $\theta$ of a neural network $J_\theta$, which assigns a differentiation potential to each cell according to its gene expression profile $x$ (Fig. 1b). Of note, this potential is not a function of space but only of gene expression. The function $J_\theta$ formalizes the Waddington epigenetic landscape, where undifferentiated cells have a high potential and, as they differentiate, move toward low-potential transcriptomic states, which correspond to mature cell types[18]. The transition to these low-potential attractor states defines a causal model of cellular dynamics capable of predicting future gene expression patterns and suggesting potential driver genes and transcriptional regulators (Fig. 1c).

STORIES's potential-based approach provides two interpretable and biologically meaningful outputs: (i) the potential $J_\theta(\mathbf{x})$, naturally orders cells $x$ along a differentiation process; (ii) the vector $-\nabla_x J_\theta(\mathbf{x})$ gives the direction of the evolution of gene expression. On the contrary, pseudotime-based methods[9,32] focus on the first aspect, and velocity-based[11,33] methods focus on the second. Crucially, STORIES also innovates compared to state-of-the-art potential-based methods[15–17] by enabling the use of spatial coordinates.

Briefly, STORIES trains the neural network $J_\theta$ by predicting a distribution $\rho_{t_k}(\theta)$ of gene expression profiles for each time point $t_k$ where $k \in [1, \ldots, K]$. These distributions are then compared to the ground-truth distributions $\mu_{t_k}$, and the parameters $\theta$ are updated to improve the predictions. Unlike existing potential-based methods, STORIES allows one to take into account the spatial coordinates of cells when comparing the distributions of gene expression.

Formally, let $\mu_t = \sum_i a_i \delta_{(x_i, r_i)}$ denote the empirical distribution of cells at time $t$, characterized by their gene expression profile $x_i \in R^d$, spatial coordinates $r_i \in \mathbb{R}^2$ and weight $a_i \in \mathbb{R}_+$ where $\sum_i a_i = 1$. Similarly, let us denote $\rho_t(\theta) = \sum_j b_j \delta_{(y_j, s_j)}$ the predictions of STORIES at time $t$. Unlike the gene expression profiles $x_i$, $y_j$, the spatial coordinates $r_i$, $s_j$ are not directly comparable because the slices are not necessarily aligned between time points. In other words, the spatial coordinates $r_i$, $s_j$ are defined up to an isometry (for example, a rotation or translation).

Existing potential-based methods train the neural network using a linear OT objective, which is sensitive to isometries. Our approach instead uses a recently developed quadratic extension of OT called FGW[23], which renders the model invariant to spatial isometries. The FGW distance, defined in equation (1), and explained more thoroughly in the Methods, allows one to compare the distributions $\mu_t$ and $\rho_t$ directly on gene expression profiles, and up to an isometry on spatial coordinates.

$$\mathrm{FGW}^\varepsilon_\alpha(\mu_t, \rho_t) = \min_{\mathbf{P} \in \mathcal{U}(\mathbf{a}, \mathbf{b})} (1 - \alpha)\mathrm{L}(\mathbf{P}) + \alpha\mathrm{Q}(\mathbf{P}) - \varepsilon\mathrm{E}(\mathbf{P}) \qquad (1)$$

FGW seeks a matrix $\mathbf{P}$ mapping cells from $\mu_t$ to $\rho_t$ such that $\mathbf{P}$ minimizes the sum of three terms: (i) the linear term L compares the gene expression coordinates $x_i$, $y_j$; (ii) the quadratic term Q compares pairwise distances $d(r_i, r_{i'})$ and $d(s_j, s_{j'})$, which are not affected by translating or rotating the tissue; (iii) an entropic regularization term E. The parameter $\alpha \in [0, 1]$ denotes the relative weight of spatial information.

Our proposed objective function evaluates the predictions across all time points using a debiased version of the FGW distance denoted $\overline{\mathcal{FG} \, \mathcal{W}^\varepsilon_\alpha}$ (Methods) as given by equation (2):

$$\mathcal{L}(\theta) = \sum_{k=1}^{K-1} (t_{k+1} - t_k)\overline{\mathcal{FG} \, \mathcal{W}^\varepsilon_\alpha}(\mu_{t_{k+1}}, \rho_{t_{k+1}}(\theta)) \qquad (2)$$

For $\alpha = 0$, equation (2) corresponds to a model relying purely on linear OT and which does not leverage spatial information, as proposed in the state-of-the-art approaches[15–17]. In the following, we refer to this as the linear method. Existing methods[15–17] propose different strategies to make the predictions $\rho_{t_{k+1}}(\theta)$. They vary in terms of teacher-forcing, number of steps between $t_k$ and $t_{k+1}$, and whether steps are implicit or explicit ('Discretization' in Methods). As explained in the following section, we thus compared those implementation choices to achieve the best performances for both STORIES and the linear method (Extended Data Fig. 1).

STORIES is implemented as an open-source Python package seamlessly integrated into the classical Python single-cell analysis pipeline[24]. Users can thus take advantage of Scverse tools like Scanpy, Squidpy and CellRank for preprocessing and downstream analysis[26,27,34]. In addition, STORIES provides a user-friendly visualization of driver genes and enriched transcription factors, thus helping biological interpretability. Finally, STORIES leverages GPU acceleration to train models efficiently (less than 20 min on a dataset of 396,000 cells and seven time points with an A40 Nvidia GPU).

In the following sections, we extensively benchmark STORIES against the available state-of-the-art methods using large-scale spatiotemporal atlases. While stVCR[22] is the only existing method using space to perform trajectory inference, no code implementation is provided to compare it with STORIES. Regarding state-of-the-art OT-based methods that do not use spatial information, most of them don't have a code implementation allowing their application on new data[14,17]. PRESCIENT[16] is thus the only state-of-the-art method that could be compared to STORIES on our benchmark datasets.

## Benchmarking STORIES against the state of the art

We assessed the effectiveness of STORIES in predicting cell states over time across three Stereo-seq spatiotemporal atlases: a mouse development atlas, a zebrafish development atlas and an axolotl brain regeneration atlas[4,6,7]. Details on data processing are provided in the Methods.

From each atlas, we created three sets: a training set, an early test set and a late test set (Fig. 2a). The test sets are composed of two time points, and the goal is to use the first time point to predict the second time point's gene expression. The late test set is particularly challenging because its second slice comes from an entirely new time point, which may contain cell states not seen during training. For example, in the zebrafish atlas, fast muscle cells only appear at 24 h post fertilization (that is, hpf), whereas the training set includes slices only up to 18 hpf.

$\alpha$ is a key parameter in STORIES as it weighs the relative importance of the spatial information in the FGW loss. To set a value for $\alpha$ in the benchmark, we want to obtain the best tradeoff between accurately predicting gene expression at the slice level and making a prediction that is consistent with space. To optimally evaluate this tradeoff, we solve an FGW problem between STORIES's gene expression predictions and the ground-truth measurements and use the linear term of the solution as a gene expression error and the quadratic term as a measure of spatial consistency (Methods). Based on this evaluation, (Extended Data Fig. 2a) setting an $\alpha$ in the range $[1 \times 10^{-3}, \ldots, 5 \times 10^{-3}]$ achieves the best tradeoff. Therefore, we use $\alpha = 5 \times 10^{-3}$ for the rest of the benchmark.

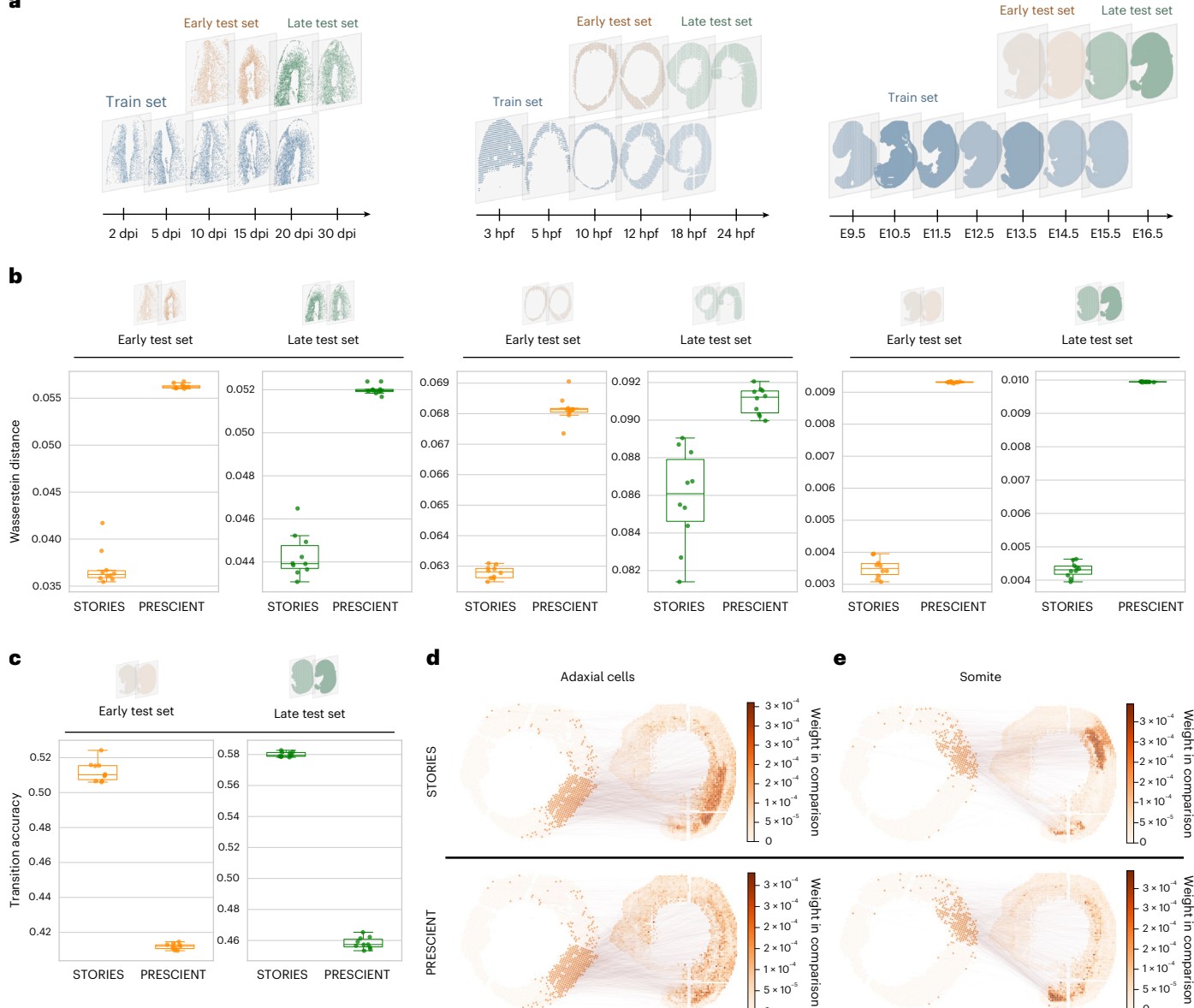

**Fig. 2 | Benchmark of STORIES on three large datasets. a**, Visual representation of the three datasets in our benchmark. From left to right: an axolotl brain regeneration dataset, zebrafish development and mouse development. Slices in each dataset are split into a train set (blue), an early test set (orange) and a late test set (green). **b**, Wasserstein distance between predicted and ground-truth gene expression in early test sets (orange) and late test sets (green) across the three datasets. Scores are reported for $n = 10$ initialization seeds. In the box plots, the center line, box limits and whiskers denote the median, upper and lower quartiles and 1.5 times the interquartile range, respectively. **c**, Cell-type transition accuracy on the mouse development dataset in the early test set and late test

set. Scores are reported for $n = 10$ initialization seeds. In the box plots, the center line, box limits and whiskers denote the median, upper and lower quartiles, and 1.5 times the interquartile range, respectively. **d**, Visual representation of the linear OT matching between the gene expression predicted by STORIES (top) or PRESCIENT (bottom) and the target gene expression measured at the next time point for adaxial cells of the zebrafish development dataset. The left slice displays the positions of cells belonging to that cell population, and the right slice displays the cells they are matched with at the following time point. **e**, Same plot as in **d**, but for somite cells of the zebrafish development dataset.

To evaluate the impact of spatial information, we compared STORIES with its linear counterpart $\alpha = 0$. Since the latter incorporates the best-performing aspects of state-of-the-art methods[15–17] for our experiments ('Discretization' in the Methods and Extended Data Fig. 1) and shares the same backbone as STORIES, it offers an unbiased way to assess the usefulness of space in trajectory inference. For this comparison, we used the classical metric in the context of trajectory inference: the Wasserstein distance[30]. This metric is only used for evaluation since it disregards the spatial information and thus would not allow us to visualize the tradeoff between spatial consistency and gene expression prediction. As shown in Extended Data Fig. 2b, adding the space

improves performance according to the Wasserstein metric in all test cases. Additionally, we displayed in Extended Data Fig. 2c–e qualitative examples showing why the training loss of STORIES is more biologically relevant than its linear counterpart. Since both methods involve matching predictions with a reference population of cells, we compared their matchings for specific cell types (Methods). First, in the axolotl atlas (Extended Data Fig. 2c), STORIES correctly matches predictions from immature neurons (IMNs) with Nptx$^+$ excitatory neurons in the lateral pallium (NptxEX), and predictions from regeneration intermediate progenitor cells (rIPC2) with excitatory neurons in the dorsal pallium (dpEX)[7]. The linear method, on the contrary, incorrectly matches rIPC2

predictions with microglial cells from a different anatomical region. Second, in the zebrafish atlas (Extended Data Fig. 2d), STORIES accurately matches predictions from the optic vesicle with cells located around the eye, and predictions from the polster with cells located within the head. In contrast, the linear method incorrectly matches optic vesicle predictions with a broad group of cells across different anatomical regions, and polster predictions with cells from the tail area. Third, in the mouse atlas (Extended Data Fig. 2e), STORIES correctly matches predictions from liver and lung cells with their respective organs. The linear method, instead, incorrectly matches lung cell predictions with a broad group of cells across organs.

We now benchmark STORIES against PRESCIENT, which is the only available state-of-the-art method for the benchmark datasets (Methods). As shown in Fig. 2b, according to the Wasserstein distance, STORIES outperforms PRESCIENT on all test cases. We then evaluated the cell-type transition accuracy of both approaches on the mouse dataset (Fig. 2c). Indeed, for mouse development, biologically valid cell-type transitions were provided by Klein D. et al.[21]. We can thus assess to which extent the predictions of STORIES and PRESCIENT fit with this ground truth (Methods). STORIES outperforms PRESCIENT also according to this evaluation metric.

Finally, in Fig. 2d,e we display practical examples where STORIES, due to its innovative usage of the space, identifies better trajectories than PRESCIENT (Methods). In zebrafish development when transitioning from 12 hpf to 18 hpf, PRESCIENT wrongly predicts Adaxial cells to be distributed all over the embryo (Fig. 2d). On the contrary, STORIES correctly matches Adaxial cells with cells close to the notochord. Indeed, Adaxial cells are known to reside next to the notochord[35]. In addition, Adaxial cells are known to differentiate into slow muscle cells[35] and indeed based on the annotation of 24 hpf (Extended Data Fig. 3a), slow muscle cells are localized in the same area where STORIES predicts Adaxial cells to evolve. In addition, in the same setting, PRESCIENT wrongly predicts somite cells to match only with the tail area missing to identify most of the somite cells annotated in the 18 hpf zebrafish embryo (Fig. 2e). On the opposite, STORIES correctly matches somite cells from 12 hpf with all somite cells from 18 hpf. See Extended Data Fig. 3b,c for other cell types in the zebrafish dataset. Finally, also in axolotl transition from 15 days post injury (dpi) to 20 dpi, we can observe a stronger mapping for Wnt+ ependymoglial cells (wntEGCs) and RrIPC1 cells in STORIES (Extended Data Fig. 4a) with respect to PRESCIENT (Extended Data Fig. 4b). In the same way, in mouse transition from embryonic day (E) 14.5 to E15.5, we observed a more accurate mapping for 'choroid plexus' and 'jaw and tooth' in STORIES (Extended Data Fig. 5) with respect to PRESCIENT (Extended Data Fig. 6).

STORIES's superior performance in achieving biologically coherent and accurate gene expression predictions demonstrates the substantial benefit of considering spatial information when learning a gradient flow model on spatial transcriptomics data.

### STORIES reveals drivers of neuron regeneration in axolotls

To further assess the potential of STORIES for trajectory inference in spatial transcriptomics through time, we first focused on axolotl brain regeneration.

We trained STORIES as described in Methods on the subset of cells described in the original publication as involved in neuron regeneration: wntEGCs and reactive ependymoglial cells (reaEGCs), regeneration intermediate progenitor cells (rIPC1 and rIPC2), IMNs, Nptx+ lateral pallium excitatory neurons (nptxEX), dorsal pallium excitatory neurons (dpEX) and medial pallium excitatory neurons (mpEX)[7]. As shown in Fig. 3a, STORIES learns an energy landscape consistent with the original publication. Indeed, the potential $J_\theta$ assigns a high potential to progenitor states (wntEGCs and reaEGCs), a medium potential to intermediary states (rIPC1, rIPC2 and IMNs) and a low potential to mature states (nptxEX, dpEX and mpEX).

We computed cell–cell transitions by applying CellRank on the gradient of the trained potential (Methods), as visualized in Fig. 3b. These transitions highlight that STORIES not only detects the correct stage of differentiation, but also recovers the three major trajectories described in the original publication: wntEGC-mpEX, reaEGC-rIPC2-dpEX and reaEGC-rIPC1-IMN-nptxEX[7]. For an overview of cell-type transitions, see Extended Data Fig. 7a. Importantly, the authors identified these trajectories by applying Monocle separately on three spatial regions and specifying EGCs as the starting point in each case. In contrast, STORIES achieves the same results without the need to isolate specific spatial regions and specify the starting point of the trajectory. Indeed, STORIES leverages spatial information to process all regions simultaneously and leverages temporal information to infer progenitor states from the data.

We then investigated how the spatial environment influences these cell trajectories. Focusing on 15 dpi, we observed that the reaEGCs have a different cell fate depending on their spatial location in the slice (Fig. 3c,d, see Extended Data Fig. 7b for the other time points and Methods for how cell fate probabilities are computed). ReaEGCs on the right of the injury tend to commit more toward mpEX, while reaEGCs on the left of the injury tend to commit more toward nptxEX. This observation seems to be in accordance with the spatial location of the terminal states in the slice. In addition, this same spatial organization of the terminal states can be observed in the additional two replicates available (Extended Data Fig. 7c), supporting its biological relevance. These results suggest that the spatial environment impacts cell fate decisions. Consequently, STORIES, integrating the spatial context, is a powerful tool for cell fate inference.

We then narrowed further into the reaEGC-rIPC1-IMN-nptxEX trajectory, which the original publication studies in most detail[7]. First, we sought to confirm expected gene trends along this trajectory. The original study suggests Vim, which encodes a critical cytoskeletal protein, as a marker of reaEGCs and Nptx1, which is involved in synaptic plasticity, as a marker of NptxEX. Accordingly, STORIES recovered a clear decreasing trend for Vim expression along differentiation and a clear increasing trend for Nptx1 expression (Fig. 3e).

Next, we performed unsupervised discovery of gene trends by fitting a spline regression model along the previously mentioned trajectory (Methods). Figure 3f reports the best candidate driver genes across differentiation stages. Interestingly, the early stages of differentiation coincide with high expression of Hes5 (Fig. 3g), which is known to maintain stemness in the context of neural differentiation[36], and Cdc25b, a cell cycle regulator key to neuron production[37,38]. Conversely, late stages of differentiation coincide with high expression of the microtubule-associated protein gene Map1a, crucial to neural development and regeneration[39], and L1cam, shown to promote axonal regeneration[40]. STORIES also outputs additional genes that represent possible drivers of neuron regeneration and would require further biological investigation. For instance, STORIES uncovered a trend for late expression of the scarcely studied Nsg2 (Fig. 3g), which is thought to be involved in synaptic function and, like Nptx1, interacts with AMPA receptors[41].

Finally, our analysis revealed possible transcriptional regulators of the differentiation process (Fig. 3h) by testing transcription factor enrichment using the curated literature-based TRRUST database (Methods). The most significantly enriched transcription factor, CTNNB1, encodes β-catenin, which has been described as an essential regulator in neuron regeneration in mouse models and in limb regeneration in axolotl[42–44]. Other top transcription factors include SP1 and MYC, described in the context of neuron regeneration and computationally retrieved in axolotl limb regeneration[45–48]. Additionally, we identify SOX6, MYCN and REST, which are not widely studied in the context of regeneration but are known regulators in development[49–51]. Interestingly, a recent study predicted REST as a regulator of neuron regeneration and validated this role in a mouse model[52].

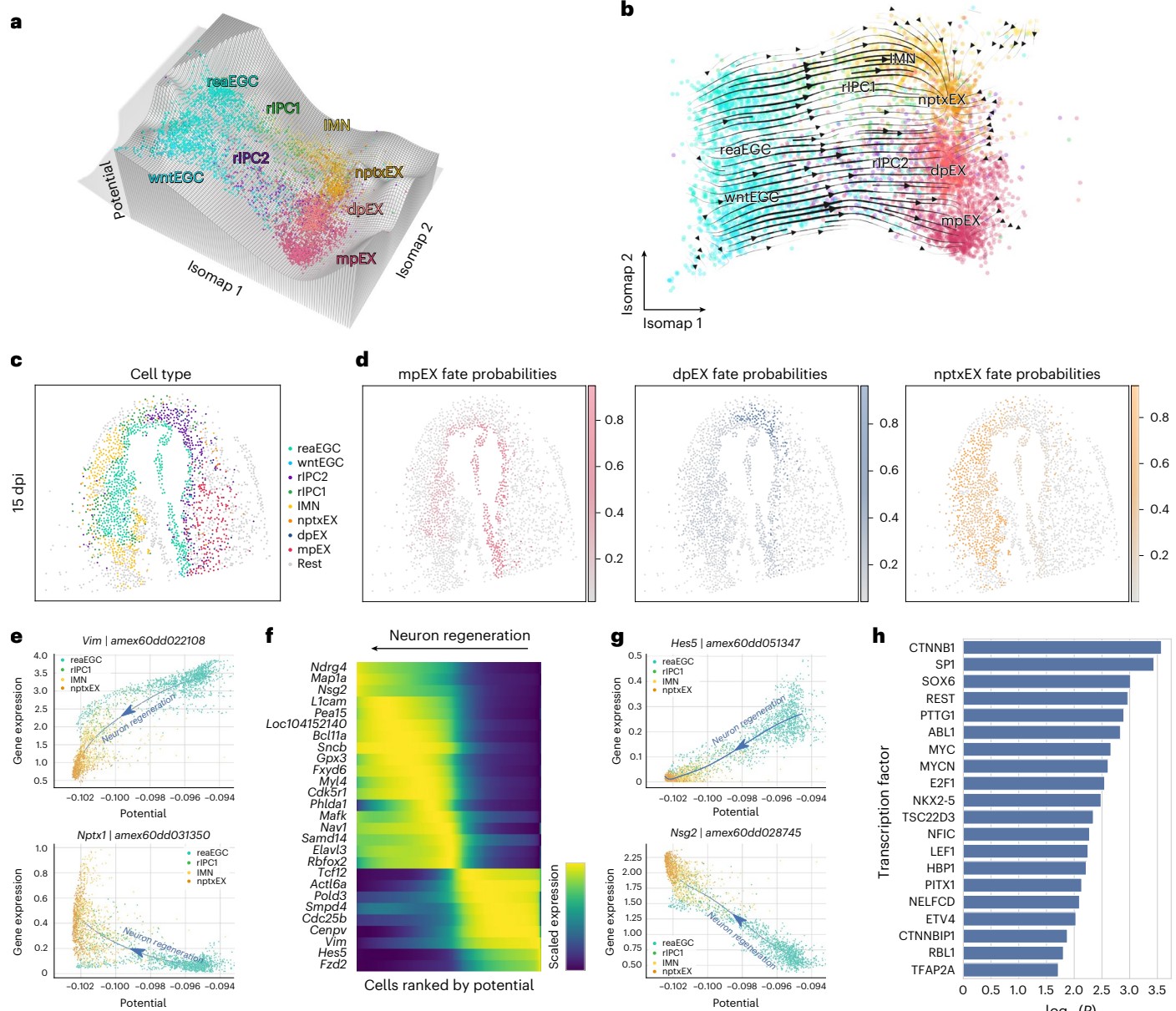

**Fig. 3 | Trajectory inference with STORIES in axolotl neuron regeneration.** **a**, Three-dimensional (3D) representation of the potential landscape learned with STORIES. The *x* and *y* axes are Isomap coordinates, and the *z* axis is an interpolation of the potential. Colors represent cell types involved in the regeneration process. **b**, Visual representation of cell–cell transitions computed using CellRank from STORIES's velocity vectors. **c**, Two-dimensional (2D) visualization of an axolotl slice at 15 dpi, with cells colored by their cell-type annotation. **d**, 2D visualization of an axolotl slice at 15 dpi, with reaEGCs colored by their predicted fate probabilities for mpEX, dpEX and npxTX fates, from left to right. **e**, Smoothed gene expression for *Vim* and *Nptx1* along the potential computed by STORIES. The blue line is a spline regression of expression from potential. **f**, Normalized gene expression regressed using a spline model along the potential computed by STORIES. Genes are ordered by the potential for which they achieve maximum expression. **g**, Smoothed gene expression for *Hes5* and *Nsg2* along the potential computed by STORIES. The blue line is a spline regression of expression from potential. **h**, Enrichment scores of transcription factors targeting candidate driver genes. A one-sided Wilcoxon rank-sum test is used to report *P* values.

Thus, not only can STORIES learn a Waddington landscape that implicitly captures the impact of space on neuron regeneration in axolotls, but it also recovers its underlying regulatory landscape through the unbiased discovery of potential drivers and mechanisms, possibly relevant for further biological investigations.

## STORIES identifies drivers of gliogenesis in mouse midbrain
We then sought to highlight STORIES's potential in trajectory inference by studying mouse dorsal midbrain development.

We trained STORIES as described in the Methods on the subset of cells described in the original article as exhibiting a branching trajectory: radial glial cells (RGCs) differentiating into either neuroblasts (NeuBs) or glioblasts (GlioBs)[4]. As visualized in Fig. 4a, STORIES learns an energy landscape consistent with the original publication. Indeed, the potential $J_\theta$ assigns a high potential to RGCs and a low potential to the more differentiated NeuBs and GlioBs.

We computed cell–cell transitions by applying CellRank on the gradient of the trained potential (Methods), as visualized in Fig. 4b. These transitions highlight that STORIES not only detects the correct stage of differentiation but also recovers the expected branching from RGC to glial and neural cell fates[4]. Importantly, the original publication identified this branching using Monocle 3, which required

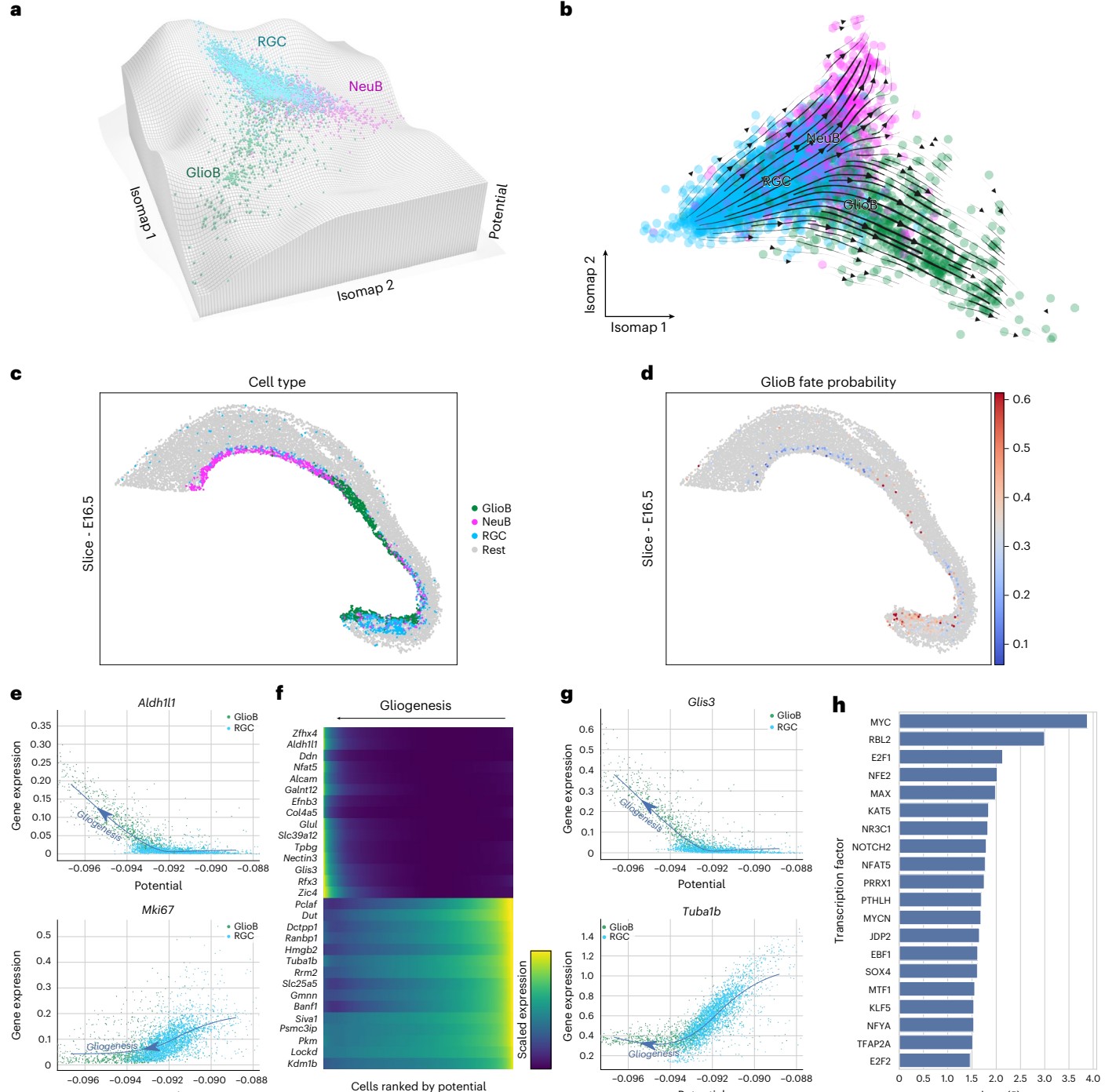

**Fig. 4 | Trajectory inference with STORIES in mouse gliogenesis. a**, 3D representation of the potential landscape learned with STORIES. The $x$ and $y$ axes are Isomap coordinates, and the $z$ axis is an interpolation of the potential. Colors represent RGCs that differentiate into either NeuBs or GlioBs. **b**, Visual representation of cell–cell transitions computed using CellRank from STORIES's velocity vectors. **c**, 2D visualization of a mouse slice at E16.5, with cells colored by their cell-type annotation. **d**, 2D visualization of a mouse midbrain slice at E16.5, with RGCs colored by their predicted probability of becoming GlioBs. **e**, Smoothed gene expression for *Mki67* and *Aldh1l1* along the potential computed by STORIES. The blue line is a spline regression of expression from potential. **f**, Normalized gene expression regressed using a spline model along the potential computed by STORIES. Genes are ordered by the potential for which they achieve maximum expression. **g**, Smoothed gene expression for *Tuba1b* and *Glis3* along the potential computed by STORIES. The blue line is a spline regression of expression from potential. **h**, Enrichment scores of transcription factors targeting candidate driver genes. A one-sided Wilcoxon rank-sum test is used to report *P* values.

manually setting RGC as the trajectories' starting point. On the contrary, STORIES achieves the same results without manual input by leveraging temporal information to infer the starting point from the data.

We then investigated how the spatial environment influences these cell trajectories. Focusing on E16.5, the differentiation of RGCs into either NeuBs or GlioBs seems to be influenced by their spatial location (Fig. 4c,d, see Extended Data Fig. 8a for the other time

points and Methods for how cell fate probabilities are computed). RGCs on the rostral part tend to commit to NeuBs, while RGCs on the extreme side of the caudal part tend to commit to GlioBs. Finally, RGCs on the central part tend to be organized into clusters of cells that commit either to NeuBs or to GlioBs. Importantly, these conclusions are supported by the agreement between the spatial position of the already differentiated cells and our predicted fate probabilities for the RGCs. These same observations apply for time point E14.5 (Extended Data Fig. 8a). In addition, this same spatial organization of the terminal states can be observed in the additional two replicates available (Extended Data Fig. 8b), supporting its biological relevance. These results suggest that the spatial environment impacts cell fate decisions. As a consequence, STORIES, integrating the spatial context, is a powerful tool for cell fate inference.

Glial cells outnumber neurons in the brain, but their development has been studied less extensively[53]. Moreover, understanding gliogenesis is of critical therapeutic importance because of its parallels with glioma, the most common and deadliest form of brain cancer[54]. Thus, we focused further on the RGC–GlioB trajectory. We first sought to confirm expected gene trends along this trajectory. The original study identifies *Mki67*, a proliferation marker, as highly expressed in RGCs, and *Aldh1l1*, an astrocyte marker, as highly expressed in GlioBs[4]. Accordingly, STORIES recovered a decreasing trend for *Mki67* expression along differentiation and an increasing trend for *Aldh1l1* expression (Fig. 4c).

Next, we performed unsupervised discovery of gene trends by fitting a spline regression model along the previously mentioned trajectory (Methods). Figure 4d reports the best candidate driver genes across differentiation stages. The early stages of differentiation coincide with a high expression of cell cycle genes *Gmnn*, *Rrm2* and *Hmgb2* (ref. 55). Additionally, we observed a high expression of the alpha-tubulin gene *Tuba1b* in the early stages of differentiation, as previously described in the developing brain[56]. Conversely, the late stages of differentiation coincide with the high expression of the glutamine synthetase gene *Glul*, a key astrocyte marker[57,58]. STORIES also outputs additional genes that represent possible drivers of gliogenesis and would require further biological investigation. For instance, *Glis3* displays an increasing trend along gliogenesis (Fig. 4e) but is little studied in this context. However, *Glis3* was recently suggested as a therapeutic target to suppress proliferation in glioma[59].

Finally, our analysis revealed candidate transcriptional regulators of the differentiation process (Fig. 4f) by testing transcription factor enrichment using the curated literature-based TRRUST database (Methods). Among the most enriched transcription factors, SOX4 and NOTCH2 have been studied in gliogenesis[60,61]. Additionally, STORIES recovers MYC, MYCN and MAX, which have been studied in the context of glioma[62].

This second experiment confirms that STORIES not only learns a Waddington landscape that implicitly captures the impact of space on gliogenesis in mice, but it also recovers its underlying regulatory landscape through the unbiased discovery of potential drivers and mechanisms that motivate further biological investigations.

## Discussion

Recent technological advances in spatial transcriptomics have enabled the tracking of gene expression at single-cell resolution in the spatial context of the tissue. Large datasets of spatial transcriptomics profiled through time give a unique opportunity to understand dynamic biological processes such as development and disease onset. However, their analysis requires trajectory inference tools tailored to the specific challenges of spatial data.

In this article, we proposed STORIES, a computational framework for trajectory inference from spatial transcriptomics profiled at several time points. STORIES enables a rich and spatially informed analysis of differentiation trajectories. To evaluate STORIES's performance, we benchmarked it against the state-of-the-art methods in three large Stereo-seq datasets and highlighted the advantage of considering spatial information in trajectory inference from time-course single-cell data. We further showcased STORIES's abilities in two concrete settings: axolotl neuron regeneration and mouse gliogenesis.

STORIES offers a model of population dynamics tailored for single-cell resolution spatial transcriptomics technologies like Stereo-seq or Visium HD. Given the fast-paced developments in spatial transcriptomics[1], the number of spatiotemporal atlases at single-cell resolution can be expected to increase steadily. At the same time, STORIES could be applied to low-resolution data (for example, 10x Visium), which have a spot size larger than the typical cell, using deconvolution techniques[63]. In addition, STORIES could be adapted to imaging-based technologies like MERFISH, which offer high resolution but can only detect a limited panel of genes[64].

STORIES provides an interpretable model of differentiation relying on a potential energy. Previous work shows that such potential landscapes arise naturally from simple gene regulatory networks[19]. However, potential energies cannot model complex gene regulatory networks, cell–cell communication or oscillations within a cell state[65]. Extensions to more complex energy functionals could thus lead to further insights into biological processes such as development or immune response. For instance, although numerically challenging because of their quadratic evaluation cost, interaction energies represent a very exciting opportunity to integrate cell–cell communication in trajectory inference models[66] to study the onset of complex diseases.

The major novelty of STORIES is its ability to learn a spatially informed potential. This methodological development is critical because dynamic processes such as development involve coordinated expression changes and tissue reorganization[67]. However, the learned potential operates only on gene expression, so it does not allow the prediction of future positions of cells. Including a spatial component in the energy function may provide a more comprehensive view of biological processes by predicting cell migration. Combining such an energy with the Gromov–Wasserstein geometry could help define more complex flows. Despite recent works[68], the extension of Wasserstein flows to Gromov–Wasserstein flows remains poorly understood theoretically since its introduction by Sturm[69] and is currently infeasible for practical implementation. Overcoming these challenges would pave the way for a more comprehensive modeling of biological processes. Relating this to existing models for morphogenesis, such as Alan Turing's reaction–diffusion model[70], is an exciting avenue for further research.

## Online content

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

## Methods

### Data collection

**Zebrafish.** For the training and validation sets, we used the five first time points shown in Fig. 2a of the original publication[6]: 3.3 hpf slice 1, 5.25 hpf slice 10, 10 hpf slice 11, 12 hpf slice 8 and 18 hpf slice 8.

For the test set, we used:

- 10 hpf slice 17 and 12 hpf slice 5 to evaluate prediction within the time range seen during training. These two slices were studied together in Supplementary Fig. 4a of the original publication[6].
- 18 hpf slice 11 and 24 hpf slice 4 to evaluate prediction outside the time range seen during training. These two slices were studied together in Supplementary Fig. 4e of the original publication[6].

Altogether, this represented 17,920 cells after preprocessing.

**Mouse.** For the training and validation sets, we used the seven first time points shown in Fig. 3a of the original publication[4]: E9.5 E1S1, E10.5 E1S1, E11.5 E1S1, E12.5 E1S1, E13.5 E1S1, E14.5 E1S1 and E15.5 E1S1.

For the test set, we used:

- E13.5 E1S2 and E14.5 E1S2 to evaluate prediction within the time range seen during training. These two slices were studied in Supplementary Fig. 2c of the original publication[4].
- E15.5 E1S2 and E16.5 E1S1 to evaluate prediction outside of the time range seen during training. The first slice was studied in Supplementary Fig. 2c and the second in Fig. 3a of the original publication[4].

Altogether, this represented 794,063 cells after preprocessing.

**Dorsal midbrain.** We retained one slice per time point. We used the three slices shown in Supplementary Fig. 7a of the original publication[4]: E12.5 (E1S3), E14.5 (E1S3) and E16.5 (E1S3). As in the original publication, we subset the analysis to the RGC, NeuB and GlioB cell types.

Altogether, this represented 4,581 cells after preprocessing. Note that the 'mouse' dataset and the 'dorsal midbrain' dataset originate from the same experiments but have different resolutions (bin 50 versus image-based segmentation), so the same neural network weights cannot be used in both cases.

**Axolotl.** *Benchmark*. For the training and validation sets, we used the five first time points shown in Fig. 3b of the original publication[7]: 2 dpi (rep1), 5 dpi (rep1), 10 dpi (rep1), 15 dpi (rep3) and 20 dpi (rep2). We manually removed spatial outliers in 10 dpi (rep1), 15 dpi (rep4) and 20 dpi (rep2). For the test set, we used:

- 10 dpi (rep2) and 15 dpi (rep1) to evaluate prediction within the time range seen during training. These two slices are shown in Supplementary Fig. 9 of the original publication[7].
- 20 dpi (rep3) and 30 dpi (rep2) to evaluate prediction outside the time range seen during training. These two slices are shown in Supplementary Fig. 9 of the original publication[7].

As in the original publication, we restricted the analysis to the dorsal part of the injured hemisphere. Altogether, this represented 22,083 cells.

*In-depth analysis.* For the analysis in 'STORIES reveals drivers of neuron regeneration in axolotls', we used slices 2 dpi (rep1), 5 dpi (rep1), 10 dpi (rep1), 15 dpi (rep3), 20 dpi (rep2) and 30 dpi (rep2). As before, we manually removed spatial outliers and restricted the analysis to the dorsal part of the injured hemisphere. As in the original publication, we subset the data to the following cell types: nptxEX, reaEGC, wntEGC, dpEX, mpEX, IMN, rIPC1 and rIPC2. This represented 5,904 cells.

### Preprocessing

All datasets were stored and handled as Numpy arrays[71] wrapped in Anndata objects[72]. For all datasets, we performed the following preprocessing steps.

**Cell and gene quality control.** Using Scanpy's 'sc.pp.filter_cells', we removed cells with less than 200 expressed genes. Then, we removed the top 0.1% of cells with the most expressed genes. Finally, we removed genes expressed in less than three cells using Scanpy's 'sc.pp.filter_genes'.

**Normalization and highly variable gene selection.** We applied Scanpy's 'Pearson residuals normalization' and selected 10,000 highly variable genes. Scanpy computed highly variable genes for each batch and merged them to avoid selecting batch-specific genes.

**Dimensionality reduction.** Using Scanpy's 'sc.tl.pca', we applied principal component analysis (PCA) to reduce the data to 50 dimensions. 'STORIES reveals drivers of neuron regeneration in axolotls' and 'STORIES identifies drivers of gliogenesis in mouse midbrain' in the Results used a subset of relevant cell types. This was done after PCA but before batch correction.

**Batch correction.** We applied Harmony on the PCA components to correct the batch effects, using Scanpy's 'sc.external.pp.harmony_integrate', a wrapper around harmonypy[73].

**Visualization.** Using Scanpy's 'sc.tl.umap', we applied uniform manifold approximation and projection (UMAP) to project the batch-corrected data into two dimensions. In 'STORIES reveals drivers of neuron regeneration in axolotls' and 'STORIES identifies drivers of gliogenesis in mouse midbrain' in the Results, we applied Isomap instead of UMAP. Indeed, we found that visually, Isomap respected cell-type transitions better than UMAP. We used scikit-learn's 'sklearn.manifold.Isomap'.

### Wasserstein gradient flow learning with a quadratic objective

**Notations.** Let us consider a time point $t \in \mathbb{R}$ and $\mu_t \in \mathscr{P}(\mathbb{R}^d)$ a discrete distribution of $n$ cells.

We denote $\mu_t = \sum_{i=1}^{n} a_i \delta_{\boldsymbol{x}_i}$ where the vector $\boldsymbol{x}_i \in \mathbb{R}^d$ represents the gene expression of the $i$-th cell, and the weights $a_i \in \mathbb{R}_+$ are such that $\sum a_i = 1$. In the following, for a function $f : \mathbb{R}^d \to \mathbb{R}^d$, we define the pushforward measure as $f_{\#}\mu_t = \sum_i a_i \delta_{f(\boldsymbol{x}_i)}$.

**Wasserstein gradient flow.** Similarly to previous works[15–17], we model the evolution of $\mu_t$ as a Wasserstein gradient flow for a potential energy. For one single cell $\boldsymbol{x}$, the Euclidean gradient flow is a $\boldsymbol{x}_t$ that verifies $\frac{d\boldsymbol{x}}{dt} = -\nabla J(\boldsymbol{x})$, the continuous counterpart of gradient descent. Wasserstein gradient flows extend this to the space of measures[74].

We say that $\mu_t$ is a Wasserstein gradient flow for the energy $\mathscr{F} : \mu \mapsto \sum_i J(\boldsymbol{x}_i) d\mu(\boldsymbol{x}_i) \in \mathbb{R}$ if its density verifies the continuity equation as given by equation (3):

$$\frac{\partial \mu(\boldsymbol{x}, t)}{\partial t} = \mathrm{div}(\mu(\boldsymbol{x}, t)\nabla J(\boldsymbol{x})) \qquad (3)$$

As in previous works[15–17], we do not know the potential $J : \mathbb{R}^d \to \mathbb{R}$ a priori and aim instead to learn a neural network $J_\theta$ from snapshots $\mu_{t_k} \in \mathscr{P}(\mathbb{R}^d)$ for $t_1 < \ldots < t_k < \ldots < t_K$. See 'Neural network architecture' for details about $J_\theta$.

**Discretization.** Our approach boils down to learning $J_\theta$ such that for given parameters $\theta$ and an initial population $\rho_{t_1} = \mu_{t_1}$, the predicted populations $\rho_{t_k}$ are close to the observed snapshots $\mu_{t_k}$.

To make these predictions, existing potential-based methods[15–17] differ in three main aspects: number of steps, teacher-forcing and

discretization scheme. As detailed below, we select the best-performing choices for these three aspects, as measured by validation loss in the zebrafish atlas. The linear method ($\alpha = 0$) thus differs from existing works by combining their best-performing aspects.

*Number of steps.* Hashimoto et al.[16] and Yeo et al.[16] make intermediary predictions between $\rho_{t_k}$ and $\rho_{t_{k+1}}$, that is, $\tau \ll (t_{k+1} - t_k)$. Bunne et al.[18] perform a single step instead, that is, $\tau = (t_{k+1} - t_k)$[17]. In our experiments, multiple steps did not improve results (Extended Data Fig. 1a), so we chose the computationally less expensive single-step method.

*Teacher-forcing.* Hashimoto et al.[16] and Yeo et al.[17] predict $\rho_{t_{k+1}}$ from $\rho_{t_k}$. Bunne et al.[18] introduce teacher-forcing, that is, predicting $\rho_{t_{k+1}}$ from $\mu_{t_k}$[17]. In our experiments, teacher-forcing improved results (Extended Data Fig. 1b), so we used it throughout this work.

*Discretization scheme.* To predict $\rho_{t_{k+\tau}}$ from an earlier population $\rho_t$, we used the forward Euler discretization scheme $\rho_{t+\tau} = (\text{Id} - \tau \nabla J_\theta)_{\#} \rho_t$ as Hashimoto et al.[16] and Yeo et al.[17]. In our discrete setting, this corresponds to $x_{t+\tau} = x_t - \tau \nabla J_\theta(x_t)$ for each cell $x$. Bunne et al.[18] propose using a backward Euler scheme to improve stability for large $\tau$[17]. We implemented this approach using the JAXopt package[75]. However, we did not find this to improve results in our experiments (Extended Data Fig. 1c), so we used the computationally less expensive forward method.

**Pairwise information.** Our setting differs from previous works[15–17] in that for each cell $x \in \mathbb{R}^d$ we have access to spatial coordinates $r \in \mathbb{R}^2$. Spatial coordinates are defined up to an isometry. For instance, a slice of an embryo may be rotated without changing the problem. Consequently, one cannot simply concatenate $x$ and $r$ to leverage the spatial coordinates. The next paragraph details how we used the coordinates $r$ to inform the problem, while still defining $J_\theta : x \in \mathbb{R}^d \mapsto J_\theta(x) \in \mathbb{R}$ only on gene expression.

**Learning the potential.** For each $k \in [1, ..., K]$, we compared the prediction $\rho_{t_k}$ to the reference snapshot $\mu_{t_k}$.

*Linear model.* Let us consider two discrete probability distributions $\mu = \sum_i a_i \delta_{x_i}$ and $\rho = \sum_j b_j \delta_{y_j} \in \mathcal{P}(\mathbb{R}^d)$. To compare $\mu$ and $\rho$, previous works[15–17] use the Sinkhorn divergence[76], as defined in equation (4):

$$\overline{\mathcal{W}_\varepsilon}(\mu, \rho) = \mathcal{W}_\varepsilon(\mu, \rho) - \frac{1}{2}(\mathcal{W}_\varepsilon(\mu, \mu) + \mathcal{W}_\varepsilon(\rho, \rho)), \quad (4)$$

where $\mathcal{W}_\varepsilon$ is the entropy-regularized OT[77] (entropy-regularized OT), as defined in equation (5):

$$\mathcal{W}_\varepsilon(\mu, \rho) = \min_{P \in \mathcal{U}(a,b)} \sum_{i,j} |x_i - y_j|_2^2 P_{i,j} - \varepsilon E(P). \quad (5)$$

Here, $\mathcal{U}(a, b) := \{P \in \mathbb{R}_+^{n \times m}, |, \sum_j P_{ij} = a_i \text{ and } \sum_i P_{ij} = b_j\}$ and $E(P) = -\sum_{i,j} P_{i,j}(\log P_{i,j} - 1)$ is the Shannon entropy. $\overline{\mathcal{W}_\varepsilon}$ is a debiased version of $\mathcal{W}_\varepsilon$, such that $\forall \rho, \bar{\mathcal{W}}_\varepsilon(\rho, \rho) = 0$.

*Quadratic model.* The Sinkhorn divergence between $\rho$ and $\mu$ only compares distributions of gene expression. Instead, we propose a debiased FGW loss to enforce the spatial coherence of the predictions. Let us consider $\mu = \sum_i a_i \delta_{(x_i, r_i)}$ and $\rho = \sum_j b_j \delta_{(y_j, s_j)} \in \mathcal{P}(\mathbb{R}^d \times \mathbb{R}^2)$. Gromov–Wasserstein is a quadratic extension of OT well suited to compare measures defined up to an isometry[78]. A debiased version of Gromov–Wasserstein has been used to learn a generative adversarial network (GAN)[79]. FGW[23] combines a linear and a quadratic OT term. In our setting, it is natural to use the linear term for gene expression and the quadratic term for spatial coordinates as given by equations (6) and (7):

$$\mathcal{FGW}_\alpha^\varepsilon(\mu, \rho) = \min_{P \in \mathcal{U}(a,b)} (1 - \alpha)L(P) + \alpha Q(P) - \varepsilon E(P), \quad (6)$$

where $L(P) = \sum_{i,j} |x_i - y_j|_2^2 P_{i,j}$

$$Q(P) = \sum_{i,j,i',j'} \left\| |r_i - r_{i'}|_2^2 - |s_j - s_{j'}|_2^2 \right\|^2 P_{i,j} P_{i',j'}. \quad (7)$$

Analogously to Bunne et al.[18], we introduce a debiased FGW to ensure the loss vanishes for an exact match[79], as given by equation (8):

$$\overline{\mathcal{FGW}_\alpha^\varepsilon}(\mu, \rho) = \mathcal{FGW}_\alpha^\varepsilon(\mu, \rho) - \frac{1}{2}(\mathcal{FGW}_\alpha^\varepsilon(\mu, \mu) + \mathcal{FGW}_\alpha^\varepsilon(\rho, \rho)) \quad (8)$$

A weight $\alpha = 1$ corresponds to the debiased entropy-regularized Gromov–Wasserstein[80]. As $\alpha = 0$, we recover the Sinkhorn divergence $\bar{\mathcal{W}}_\varepsilon$.

*Final loss function.* For $\alpha \in [0, 1]$, the full objective is given by equation (2):

$$\mathcal{L}(\theta) = \sum_{k=2}^{K} (t_k - t_{k-1}) \overline{\mathcal{FG}\,\mathcal{W}_\alpha}(\mu_{t_k}, \rho_{t_k}) \quad (9)$$

where $\rho_{t_k} = (f_\theta)_{\#} \mu_{t_{k-1}}$ with $f_\theta : (x, r) \mapsto (x - \tau \nabla J_\theta(x), r)$. In other words, gene expression is predicted with teacher-forcing and a single forward step ('Discretization'). Our model doesn't predict spatial coordinates; therefore, $s$ and $r$ are identical in our FGW loss and correspond to the spatial positions of cells at time $t_{k-1}$. This loss is optimized using mini-batches, as done by Yeo et al.[17] and Bunne et al.[18] A theoretical study of mini-batch OT for debiased linear and quadratic OT is provided by Fatras et al.[81].

**Choice of the quadratic weight.** To investigate the effect of the relative weight of the linear term in FGW, we reported results for $\alpha \in \{1 \times 10^{-5}, 1 \times 10^{-4}, 1 \times 10^{-3}, 5 \times 10^{-3}, 1 \times 10^{-2}, 1 \times 10^{-1}\}$. Choosing $\alpha = 1$ would not allow learning the potential $J_\theta$ because it would ignore gene expression. As expected, with a low weight ($\alpha = 10^{-5}$), STORIES behaves as the linear method (see 'Benchmarking STORIES against the state of the art' in the Results).

To compare our approach with the linear model proposed in previous works[15–17], we also trained the model with a Sinkhorn divergence, that is $\alpha = 0$.

The benchmark in 'Benchmarking STORIES against the state of the art' in the Results suggests good performances for values of $\alpha$ of the order of $10^{-3}$, with $\alpha = 5 \times 10^{-3}$ performing best. The value of $\alpha$ can be adjusted by the user depending on the dataset. In 'STORIES reveals drivers of neuron regeneration in axolotls' and 'STORIES identifies drivers of gliogenesis in mouse midbrain' in the Results, we set a value of $\alpha = 10^{-3}$.

**Computational OT**

**OTT solvers.** We use the OTT package to solve OT problems in a fast, GPU-enabled and differentiable manner[29]. In particular, we rely on the Sinkhorn and Gromov–Wasserstein solvers. We set the entropic regularization $\epsilon = 0.01$.

**Linear term: gene expression.** The linear OT terms are defined on gene expression space, for which we chose the $d$ first components of the Harmony-aligned PCA. We chose $d = 50$ as it is the one that maximizes cell-type transition accuracy (see 'Evaluation' for details on how this is computed) without increasing the computational runtime of the method (Extended Data Fig. 9a,b). Notably, in both 'STORIES reveals drivers of neuron regeneration in axolotls' and 'STORIES identifies drivers of gliogenesis in mouse midbrain' in the Results, we only used

$d = 20$ as the datasets of the case studies were smaller than in the benchmark and the analysis was conducted on a subset of the cell types. Before training the neural network, we normalized the points as $x_i \leftarrow \frac{x_i}{\max_j \|x_j\|_\infty}$ to make the linear and quadratic terms comparable.

**Quadratic term: spatial coordinates.** Before training the neural network and for each slice separately, we centered the spatial coordinates and scaled them to unit variance to make linear and quadratic terms comparable.

**Neural network architecture.** We implemented a multilayer perceptron with two hidden layers of dimension 128 and GeLU activations using Flax[82]. This is a similar architecture to previous works[15–17]. The linear output layer has no bias since it would not influence the values of $\nabla J_\theta$. Likewise, a soft activation like GeLU is preferable to the classical ReLU because we manipulate $\nabla J_\theta$. Indeed, the derivative of ReLU is simply a unit step function, which is discontinuous and not very expressive.

## Neural network training

**Data loading.** For each time point described in 'Data collection', 75% of the cells were used as training samples and 25% as validation. At each training or validation iteration, a batch containing 1,000 cells per time point was sampled uniformly without replacement. In development, the early time points contain fewer cells than the later time points. If less than 1,000 cells were available for a time point, we used all available cells. To reflect the train/validation split, one in four iterations performs a validation step.

**Optimizer.** We used Optax's implementation of the AdamW optimizer, with parameters b1 = 0.9, b2 = 0.999, eps = $1 \times 10^{-8}$ and weight_decay = $1 \times 10^{-4}$ (ref. 28). We set the learning rate using Optax's cosine scheduler, with an initial value of $1 \times 10^{-2}$ and 10,000 decay steps. To ensure convergence, when performing ten steps, we set the learning rate to $1 \times 10^{-3}$. Similarly, when performing an implicit step, we set the learning rate to $1 \times 10^{-4}$.

**Early stopping.** We set the maximum number of iterations to 15,000 but stopped the training when the validation loss had not improved in 150 iterations. We kept the weights associated with the lowest validation loss and used the orbax package to save and load checkpoints of the model.

**Seeds.** We ran every experiment with ten random seeds: 17,158; 20,181; 12,409; 5,360; 21,712; 21,781; 24,802; 13,630; 9,668; and 651. The random seed reproducibly determines the train/validation split and weight initialization. For the plots in Fig. 2c, and the analysis in 'STORIES reveals drivers of neuron regeneration in axolotls' and 'STORIES identifies drivers of gliogenesis in mouse midbrain', the experiments correspond to the randomly chosen seed 20,181.

**Computational runtime.** Due to mini-batch sampling of the data samples, STORIES can scale to very large datasets. It also leverages GPU acceleration to speed up the training of the model. See Extended Data Fig. 9c for a comparison of the training time of STORIES for different values of the $\alpha$ parameter on the three benchmark datasets. Overall, we observe that while using a greater $\alpha$ (like 0.1) makes the training longer due to Gromov–Wasserstein being a much more computationally heavy problem than its linear counterpart, using a small alpha value leads to a more than reasonable training time. Surprisingly, in some datasets, using a low alpha leads to an even faster training than using $\alpha = 0$. While this result was not expected, we could assume that for small alpha values the regularization induced by the quadratic term in FGW makes the problem converge faster as it introduces more constraint on the solution. Training progress can be monitored using the tqdm library, which provides real-time progress bars for iterative processes.

## Running PRESCIENT

We compare STORIES to PRESCIENT, which also uses OT to learn a gene expression potential governing a causal model of differentiation. We used their open-source implementation (https://github.com/gifford-lab/prescient/) and applied PRESCIENT with all default parameters except for 'train_sd'. Indeed the default value of 0.5 for this parameter, which controls the strength of the random noise in their differentiation model, was too high compared to the scale of the gene expression data in the benchmark datasets. We therefore tuned this parameter for a fair evaluation and used 0.01 for 'train_sd'.

## Evaluation

For the problem of choosing the optimal $\alpha$ parameter in Extended Data Fig. 2a,b, we solve the FGW problem with $\alpha = 10^{-3}$ and $\varepsilon = 10^{-3}$ and report the terms L ($P$) and Q ($P$) separately. Since L ($P$) quantifies the error in terms of gene expression, we call this quantity the 'gene expression prediction score'. Similarly, since Q ($P$) quantifies the error in terms of spatial coordinates, we call this quantity the 'spatial coherence score'.

For the benchmark, we use the entropy-regularized Wasserstein distance and thus solve a regularized linear OT problem with $\varepsilon = 10^{-1}$ and report the linear term (without the entropy). This quantity is also called the Sinkhorn distance in the literature[77]. Benchmark plots were produced using the Seaborn package.

For the evaluation of the cell-type transition accuracy, we used the gene expression predicted by each method at time $t + 1$, then solved a regularized linear OT problem between this predicted gene expression and the true measurements at $t + 1$. The resulting transport plan was then used as a similarity matrix on which we applied ten-nearest-neighbors classification. This resulted in cell-type predictions that we compared with the ground truth. Importantly, those cell-type predictions are only based on gene expression predictions and didn't use spatial coordinates as an input, neither in the potential nor in the transport plan. The spatial information was only leveraged by STORIES during the training, and not at prediction time.

## Qualitative evaluation of trajectories

For Fig. 2d,e and Extended Data Figs. 2c–e, 3b,c and 4–6, we compared transport plans involved in STORIES and other methods that are used to match each method's predictions with the true gene expression measurements at the next time point. Nonetheless, these two experiments differ in an important way.

In Fig. 2d,e, we show on one side the FGW transport plan used to compute STORIES's training loss between the model's predictions and gene expression measurements from the next time point. On the other side we show the Wasserstein transport plan used to compute the linear version of STORIES's training loss.

In Extended Data Figs. 2c–e, 3b,c and 4–6, we compute for both STORIES and PRESCIENT, the linear OT plan between each model's predictions and the gene expression measurements at the next time point. This allows us to compare the two methods only based on their output prediction without injecting spatial information at evaluation time.

Given a transport plan for each method, we compare how cells are matched at the cell-type level. Formally, let us consider the indicator vector $a \in \mathbb{R}^n$ where $a_i = 1$ if the $i$-th cell corresponds to a given cell type, and $a_i = 0$ otherwise. The transport plan $P$ between the prediction $\rho_t$ and the ground truth $\mu_t$ is applied to the indicator, yielding a vector $b = P\,a$ representing the mass transported from $a$ toward each cell in the second time point. These densities are plotted on top of the spatial coordinates for Fig. 2d,e and Extended Data Figs. 2c–e, 3b,c and 4–6.

## Computational growth rate

Growth rate estimation has only been performed in the two biological test cases ('STORIES reveals drivers of neuron regeneration in axolotls' and 'STORIES identifies drivers of gliogenesis in mouse midbrain' of the results), where we analyzed a restricted number of cell types and could validate the biological coherence of the computed growth rate (see details in paragraphs below). The choice of not using growth rate estimation in the benchmark was also done to have a comparison with the state-of-the-art method that is only focused on the quality of their underlying model. Indeed, other state-of-the-art methods sometimes use alternative ways to model the growth rate[14,16] or do not consider it at all[83]. Furthermore, previous works tested gene sets to compute the growth rate in the case of mice and humans, but not of zebrafish and axolotl[13,16,21]. In addition, the growth rate estimation requires knowing a gene signature that is well established for some organisms, but might be missing for less studied ones. We thus kept uniform marginals for the benchmark in 'Benchmarking STORIES against the state of the art' in the Results.

**Weight of marginals.** When comparing a prediction $\rho_{t_k}$ to the reference snapshot $\mu_{t_k}$, Yeo et al.[16] proposes setting the weights $b_j$ of the prediction $\rho_{t_k} = \sum_j b_j \delta_{y_j}$ proportionally to a computationally derived growth rate. The motivation is that cells with a larger growth rate should be matched to more descendants. This idea was introduced in Waddington OT[13] and recently reimplemented in MOSCOT[21]. In the next paragraphs, we follow MOSCOT's implementation.

**Proliferation and apoptosis.** Computing the growth rate relies on cell-wise proliferation and apoptosis scores. We used Scanpy's 'sc.tl.score_genes' with lists of genes collected from the literature. The gene lists are described in more detail at the end of this section. We calculated gene scores on raw counts, after quality filtering of cells and genes.

**Calculating the growth rate.** For a given cell $x$, let us call $prol_i \in \mathbb{R}$ the proliferation score and $apo_i \in \mathbb{R}$ the apoptosis score. We then define the birth rate $\beta_i \in \mathbb{R}$ as given by equation (10):

$$\beta_i = \beta_{min} + \frac{\beta_{max} - \beta_{min}}{1 + \exp\left(-4.0 \times \frac{prol_i - \beta_{center}}{\beta_{width}}\right)} \quad (10)$$

and the death rate $\gamma_i \in \mathbb{R}$ as given by equation (11):

$$\gamma_i = \gamma_{min} + \frac{\gamma_{max} - \gamma_{min}}{1 + \exp\left(-4.0 \times \frac{apo_i - \gamma_{center}}{\gamma_{width}}\right)} \quad (11)$$

We used MOSCOT's default parameters $\beta_{min} = \gamma_{min} = 0.3$, $\beta_{max} = \gamma_{max} = 1.7$, $\beta_{center} = 0.25$, $\beta_{width} = 0.5$, $\gamma_{center} = 0.1$ and $\gamma_{width} = 0.2$. Finally, we defined the cell's growth rate as given by equation (12):

$$g_i = \exp\left(\Delta t \times (\beta_i - \gamma_i)\right) \quad (12)$$

where $\Delta t$ is the time difference between populations. We obtained $a_i$ by normalizing the growth rate as given by equation (13):

$$a_i = \frac{g_i}{\sum_j g_j}, \text{ i.e. } \boldsymbol{a} = \text{softmax}\left(\Delta t \times (\boldsymbol{\beta} - \boldsymbol{\gamma})\right). \quad (13)$$

In this equation, $\Delta t$ plays the role of the Softmax's inverse temperature. The histogram $\boldsymbol{a}$ will thus be sharper for large values of $\Delta t$. Most slices in our experiments were evenly sampled, so we set a fixed $\Delta t = 1$, which yielded sharp enough weight differences between cell types. In this estimation, $\Delta t$ plays the role of an inverse temperature parameter. A higher $\Delta t$ will result in a more entropic distribution of

growth rates without changing the ranking of the cells in this distribution. On the other hand, using $\Delta t = 0$ leads to estimating a constant growth rate for all cells, which is therefore equivalent to not modeling growth. The growth rate models the potential division (rate > 1) or death (rate < 1) of cells between the measured time points. We relied on prior knowledge of the length of the cell cycle in the biological conditions we studied to make sure that our choice of $\Delta t$ was reasonable. In the axolotl regeneration case, for all tested values of $\Delta t$, our estimated growth rates were higher for less differentiated cells, which makes biological sense. In addition, we found in the literature that in the context of an injury, axolotl spinal cord cells might take between approximately 5 and 15 days to divide[84]. This means that cells could divide at most twice between consecutive time points in our dataset, thus giving birth to four descendants. As shown in Extended Data Fig. 10a–c, using $\Delta t = 1$ is coherent with this estimation as the maximum estimated growth rate is 4 in this case. While using a higher value for $\Delta t$ leads to unreasonably high estimated growth rates, using a value between 0 and 1 interpolates between not modeling growth and the upper bound on maximum growth between measured time points. In mouse midbrain development, we also observed that our estimated growth rates were higher for the less differentiated cells: RGCs. Moreover, we found in the literature that mouse cells between E13 and E17 (in the same range as our dataset, which has three time points at E12.5, E14.5 and E16.5) take between approximately 16 h and 26 h to divide[85]. This means that cells could replicate at most three times between consecutive time points in our dataset (which would result in a cell giving birth to eight descendants). As shown in Extended Data Fig. 10d, our estimated growth rates with $\Delta t = 1$ are coherent since they are below the theoretical upper bound.

**Dorsal midbrain.** For 'STORIES reveals drivers of neuron regeneration in axolotls', we used murine proliferation and apoptosis gene sets from MOSCOT. Proliferation genes come from ref. 86 and apoptosis genes from https://www.gsea-msigdb.org/gsea/msigdb/cards/HALLMARK_P53_PATHWAY/. See Supplementary Texts 1 and 2 for the gene names.

**Axolotl neuron regeneration.** For 'STORIES identifies drivers of gliogenesis in mouse midbrain', we used a neural stem cell (NSC) axolotl gene set described in the original publication[7] to represent proliferation and a human apoptosis gene set from MOSCOT, originally from gsea-msigdb's HALLMARK_APOPTOSIS. See Supplementary Texts 3 and 4 for the gene names.

## Trajectory inference

**Gene imputation.** In our analysis, the gene expression trends (Figs. 3 and 4) would be negatively affected by the sparsity of gene expression. CellRank demonstrated good performances in identifying gene expression trends with MAGIC[27,87]. We thus applied MAGIC gene imputation after all other preprocessing steps. We computed the exponentiated Markov transition matrix on the Harmony-aligned PCA space instead of the original PCA. We did not use the imputed signal for tasks other than gene expression trends.

**Potential visualization.** The neural potential $J_\theta$ is a functional defined on the $d$-dimensional space of Harmony-aligned principal components. To visualize the potential as a Waddington-like landscape defined on two dimensions, we proceed similarly to Qin et al.[88].

- First, we compute the potential $J_\theta(x)$ associated with each cell $x$.
- Then, we use Scipy's RBF interpolation and the 2D Isomap coordinates of the cells to define a potential on a 2D grid[89].
- The cells are projected on the surface using the interpolator.
- Finally, the maximum value is thresholded.

We rendered the resulting surface and point cloud using Blender's Python API.

**Cell–cell transition matrix.** We used CellRank's VelocityKernel with a velocity $v(x) = -\nabla J_\theta(x)$ for a trained $J_\theta$. Based on this kernel, we computed a cell–cell transition matrix, and the trajectory plots in Figs. 3b and 4b.

**Cell fate probabilities.** After computing the VelocityKernel, we use CellRank's GPCCA estimator to compute cell fate probabilities. We set the number of macrostates to the number of cell types present in the dataset and set the terminal states to ['dpEX', 'nptxEX', 'mpEX'] for the axoltol and ['neuB', 'glioB'] for the mouse midbrain. Extended Data Fig. 7a was obtained by aggregating those cell fate probabilities at the cell-type level with the CellRank function 'cellrank.pl.aggregate_fate_probabilities'.

**Gene expression trends.** We fit the MAGIC-imputed gene expression as a function of the learned potential, using Scipy's Spline Regression[89]. To plot the gene expression cascades in Figs. 3d and 4d, we order genes by the value of potential for which the maximum regressed expression value is achieved. Genes are then split into equally sized groups illustrating different stages of differentiation (10 groups for Fig. 3d and 2 groups for Fig. 4d). Finally, regressed values for the genes with the best regression scores in each group are displayed (3 per group in Fig. 3d and 15 per group in Fig. 4d).

**Transcription factor enrichment.** We perform transcription factor-target enrichment based on the TRRUST dataset, which contains 'activation', 'repression' and 'unknown' links based on curated literature. For each transcription factor in the database, we perform a one-sided Wilcoxon rank-sum test comparing the list of regression scores of its $n_1$ target genes, and the list of regression scores of the other $n_2$ genes. The number of target genes is different for each transcription factor, but in all cases $n_1 + n_2 = 10,000$. Figures 3h and 4h display the transcription factors ranked by $P$ value.

### Reporting summary

Further information on research design is available in the Nature Portfolio Reporting Summary linked to this article.

### Data availability

We retrieved the mouse Stereo-seq atlas from Chen et al.[4], available at https://db.cngb.org/stomics/mosta/. We retrieved the zebrafish Stereo-seq atlas from Liu et al.[6], available at https://db.cngb.org/stomics/zesta/. We retrieved the axolotl Stereo-seq atlas from Wei et al.[7], available at https://db.cngb.org/stomics/artista/. We retrieved mouse proliferation genes from Tirosh et al.[86] (Supplementary Text 1) and apoptosis genes from gsea-msigdb's HALLMARK_P53_PATHWAY (Supplementary Text 2), available at https://www.gsea-msigdb.org/gsea/msigdb/human/geneset/HALLMARK_P53_PATHWAY.html. We retrieved an NSC axolotl gene set from Wei et al.[7] (Supplementary Text 3) and a human apoptosis gene set from gsea-msigdb's HALLMARK_APOPTOSIS (Supplementary Text 4), available at https://www.gsea-msigdb.org/gsea/msigdb/human/geneset/HALLMARK_APOPTOSIS.html. Source data are provided with this paper.

### Code availability

The Python package for STORIES is hosted at https://github.com/cantinilab/stories/ (ref. 24). It can be installed easily by running 'pip install stories-jax'. Code to reproduce the experiments and figures is available at https://github.com/cantinilab/stories_reproducibility/.

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

### Acknowledgements

The project leading to this manuscript has received funding from the European Union, European Research Council StG and MULTIview-CELL (101115618, to L.C.). In addition, this work has been funded by the French government under management of Agence Nationale de la Recherche as part of the 'Investissements d'avenir' program (ref. ANR-19-P3IA-0001; PRAIRIE 3IA Institute; to L.C. and G.P.) and by the Inception program 'Investissement d'Avenir grant ANR-16-CONV-0005' (to L.C.). The work of G.P. was supported by the European Research Council (project NORIA). This work was performed using HPC resources from GENCI–IDRIS (grant no. 2024-AD011013214R3). We acknowledge the help of the HPC Core Facility of the Institut Pasteur and D. Philipps for administrative support.

## Author contributions

G.-J.H., G.P. and L.C. designed and planned the study. G.-J.H. developed the tool. G.-J.H. and J.S. performed most analyses. A.A. contributed to the growth rate analysis. D.C. helped with discussing the impact of space on cell fate trajectories. G.-J.H., J.S. and L.C. wrote the paper. G.P. revised the manuscript. All authors read and approved the final manuscript.

## Competing interests

The authors declare no competing interests.

## Additional information

**Extended data** is available for this paper at https://doi.org/10.1038/s41592-025-02855-4.

**Correspondence and requests for materials** should be addressed to Laura Cantini.

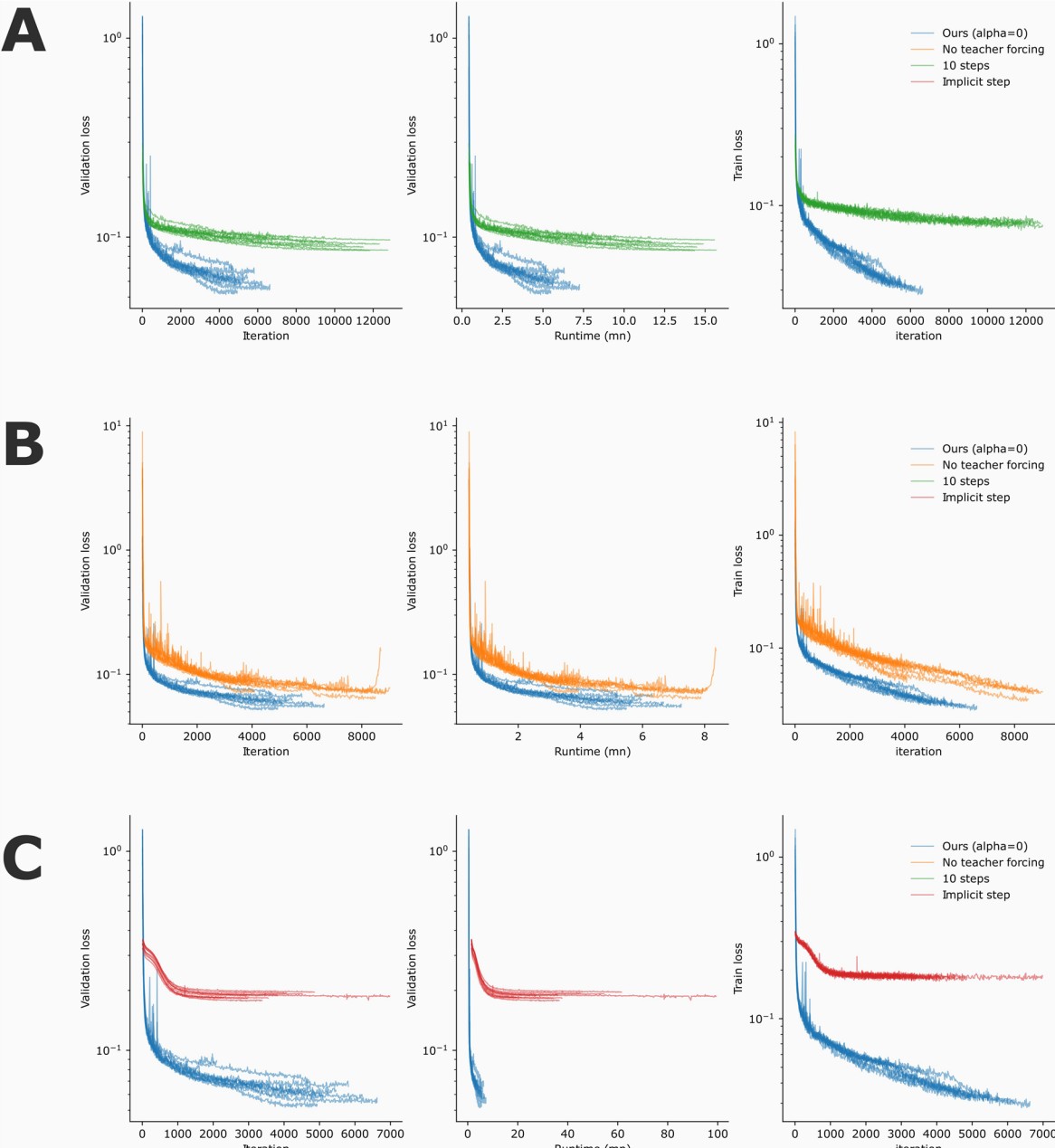

**Extended Data Fig. 1 | Comparison of different choices of discretization during training on the zebrafish atlas.** All methods are run for 10 different initialization seeds. (**A**) Training and validation losses along iterations and runtime, for the linear method (blue) and a version with 10 forward steps (green), as in [Hashimoto et al., 2016]; (**B**) Training and validation losses along iterations and runtime, for the linear method (blue) and a version without teacher forcing (orange), as in [Hashimoto et al., 2016]; (**C**) Training and validation losses along iterations and runtime, for the linear method (blue) and a version with an ICNN implicit step (red), as in [Bunne et al., 2022].

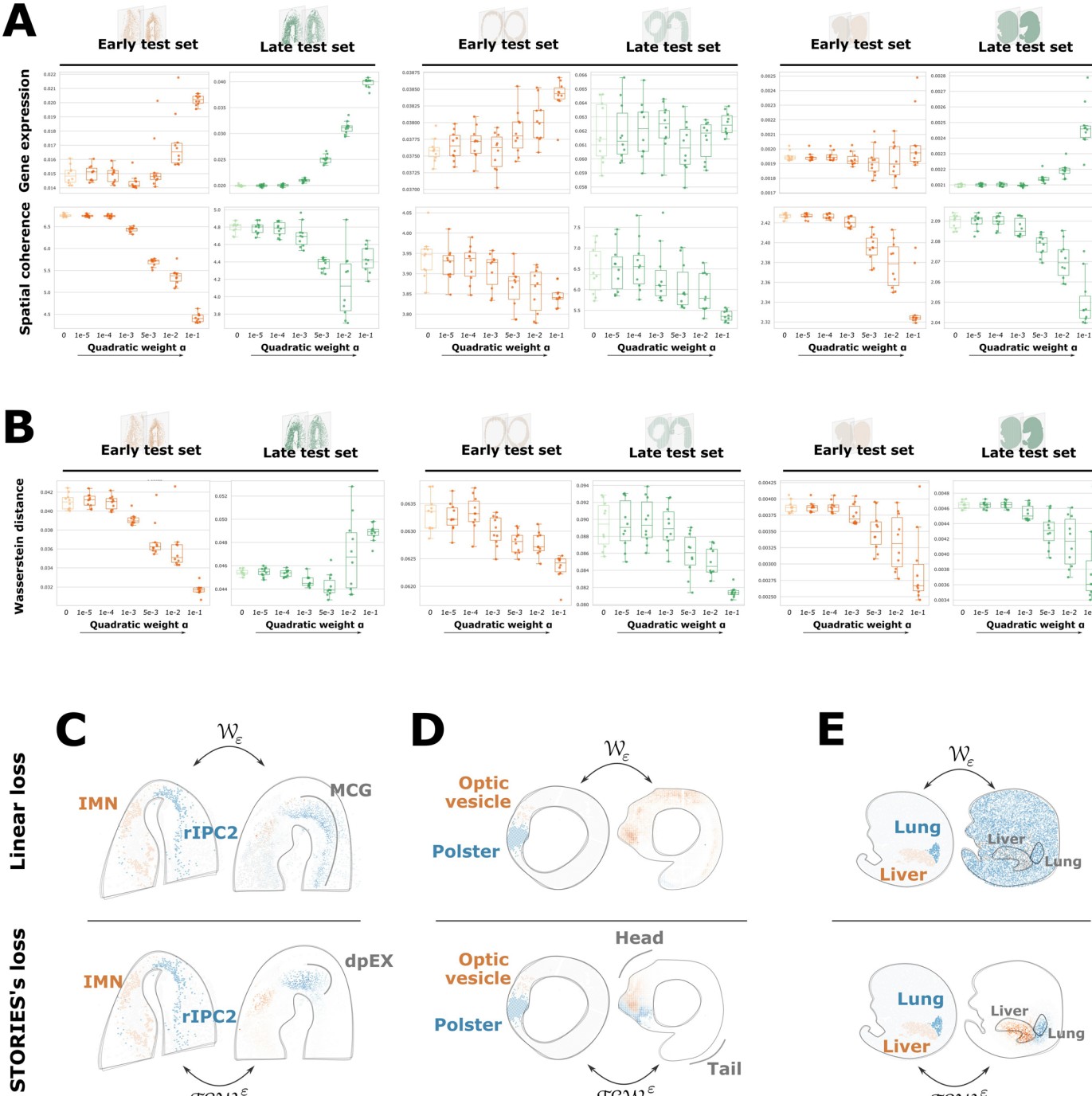

**Extended Data Fig. 2 | Comparison between STORIES and its linear counterpart.** (**A**) In the boxplots, the center line, box limits, and whiskers denote the median, upper and lower quartiles, and 1.5× interquartile range, respectively. All scores are reported for n = 10 initialization seeds. Gene expression prediction score (top) and spatial coherence score (bottom) in early test sets (orange) and late test sets (green) across the three datasets. Scores are reported across seven values of quadratic weight α parameter, including the linear method (α = 0, light orange/green); (**B**) Wasserstein distance score in early test sets (orange) and late test sets (green) across the three datasets. Scores are reported across seven values of quadratic weight parameter α, including the linear method (α = 0, light orange/green); (C-E) Visual representation of the optimal transport matching involved in the loss of the linear method (top) and STORIES (bottom), for the three benchmark datasets. In each dataset, the left slice displays two cell types and the right slice displays the cells they are matched with at the following time point. The shown examples are: (**C**) IMN and rIPC2 cells in the axolotl dataset; (**D**) optic vesicle and Polster cells in the zebrafish dataset; (**E**) lung and liver cells in the mouse dataset.

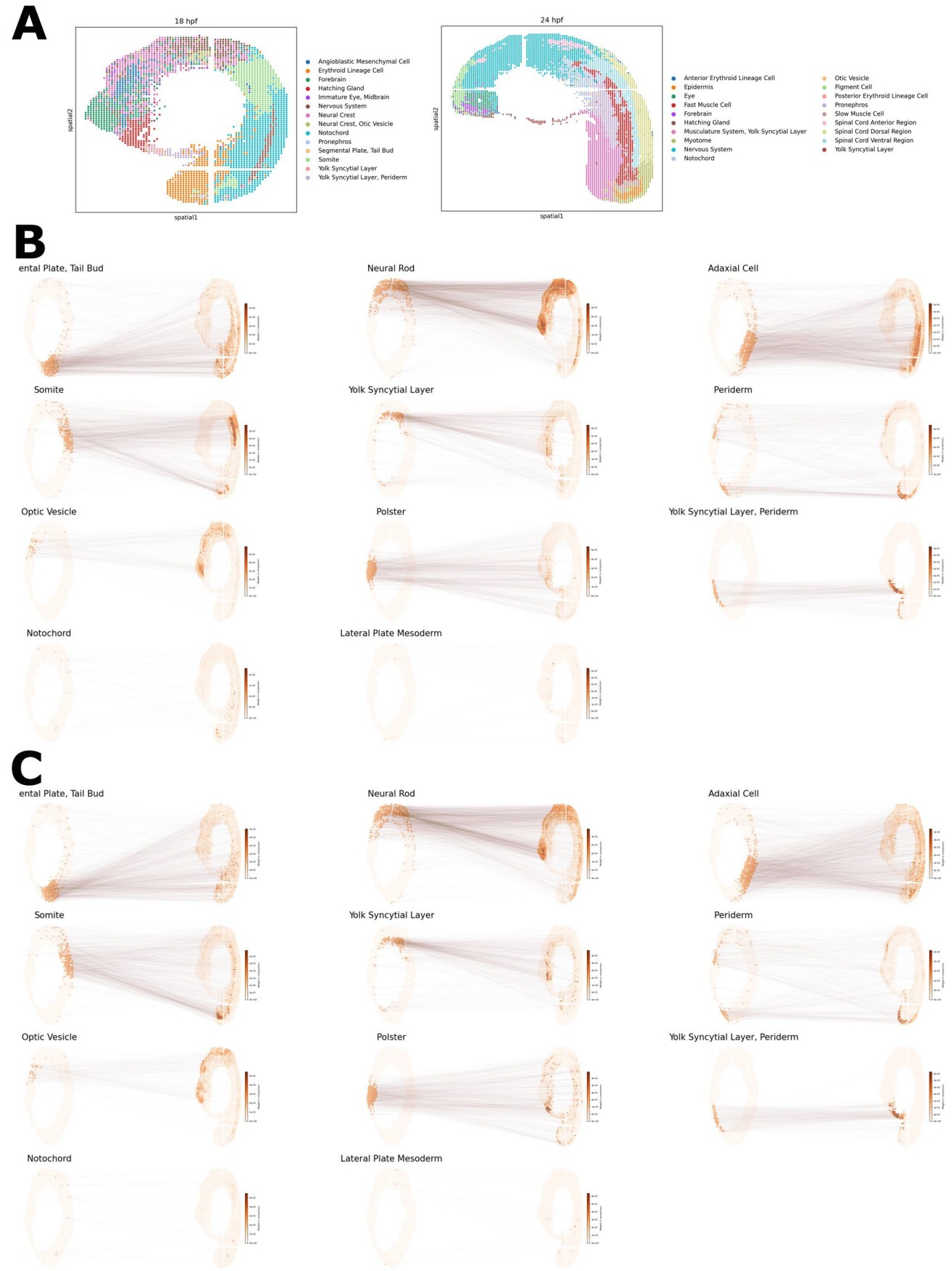

**Extended Data Fig. 3 | See next page for caption.**

**Extended Data Fig. 3 | Qualitative assessment of gene expression predictions in zebrafish development.** (**A**) Zebrafish slices at time points 18hpf and 24hpf with cells colored by their cell type annotations; (**B**, **C**) Visual representation of the linear optimal transport matching between the gene expression predicted by each method (STORIES in (**B**) and PRESCIENT in (**C**)) and the target gene expression measured at the next time point for each cell type present at this time point of the zebrafish development dataset. For each example, the left slice displays the positions of cells belonging to that cell population and the right slice displays the cells they are matched with at the following time point.

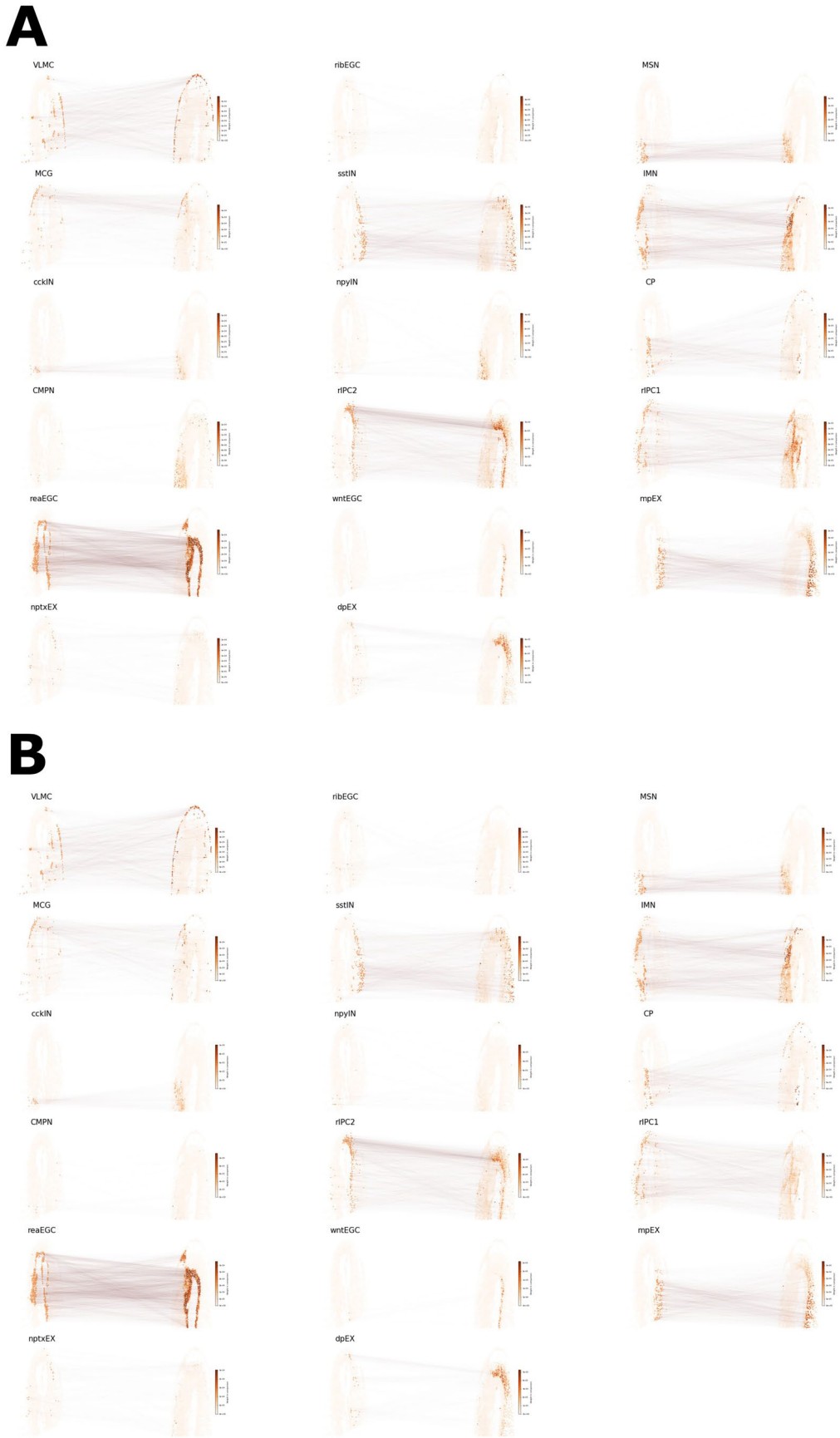

**Extended Data Fig. 4 | Qualitative assessment of gene expression predictions in axolotl regeneration.** Visual representation of the linear optimal transport matching between the gene expression predicted by each method (STORIES in (**A**) and PRESCIENT in (**B**)) and the target gene expression measured at the next time point for each cell type present at this time point of the axolotl regeneration dataset. For each example, the left slice displays the positions of cells belonging to that cell population and the right slice displays the cells they are matched with at the following time point.

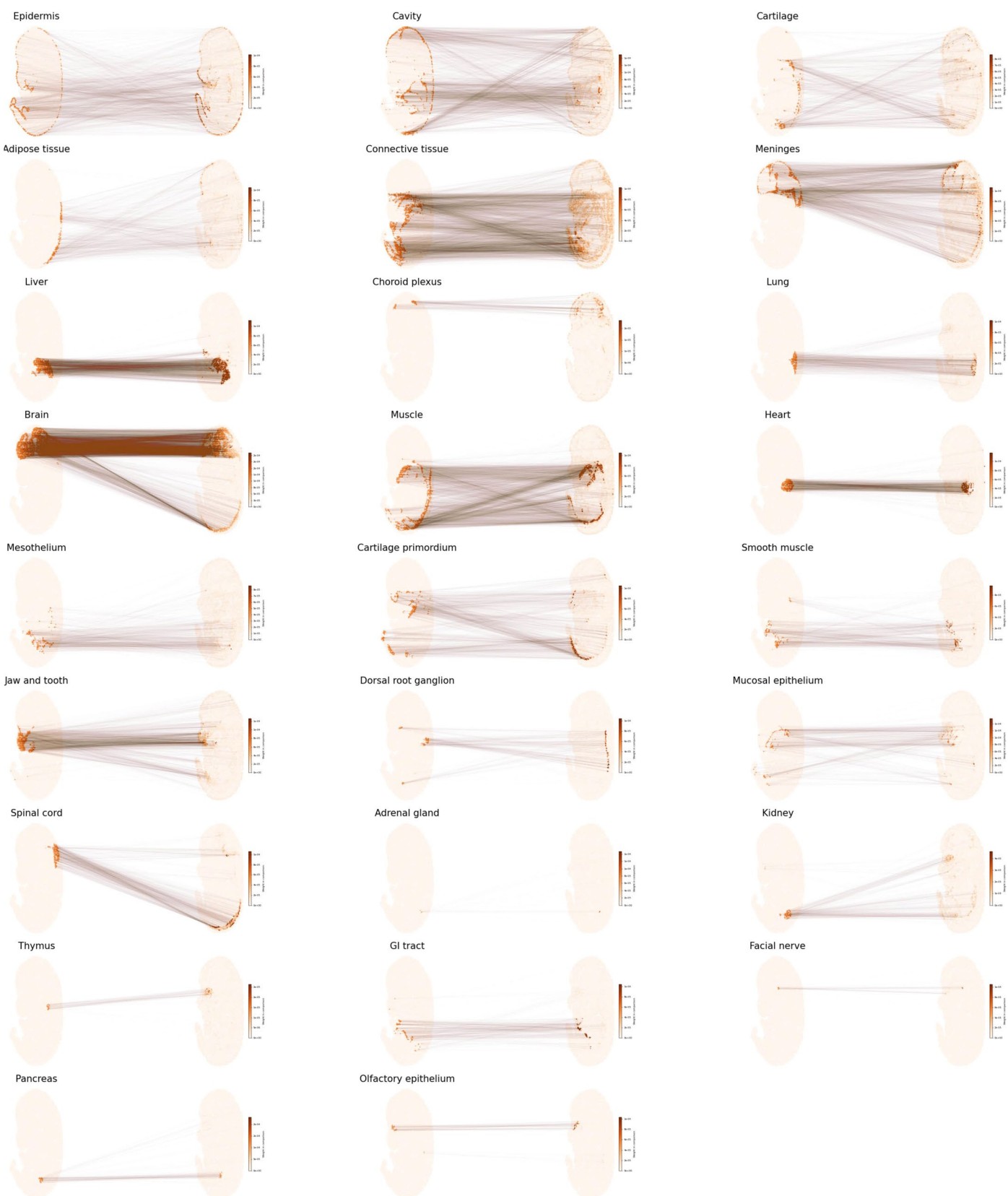

**Extended Data Fig. 5 | Qualitative assessment of STORIES' gene expression predictions in the mouse development dataset.** Visual representation of the linear optimal transport matching between the gene expression predicted by STORIES and the target gene expression measured at the next time point for each cell type present at this time point of the mouse development dataset. For each example, the left slice displays the positions of cells belonging to that cell population and the right slice displays the cells they are matched with at the following time point.

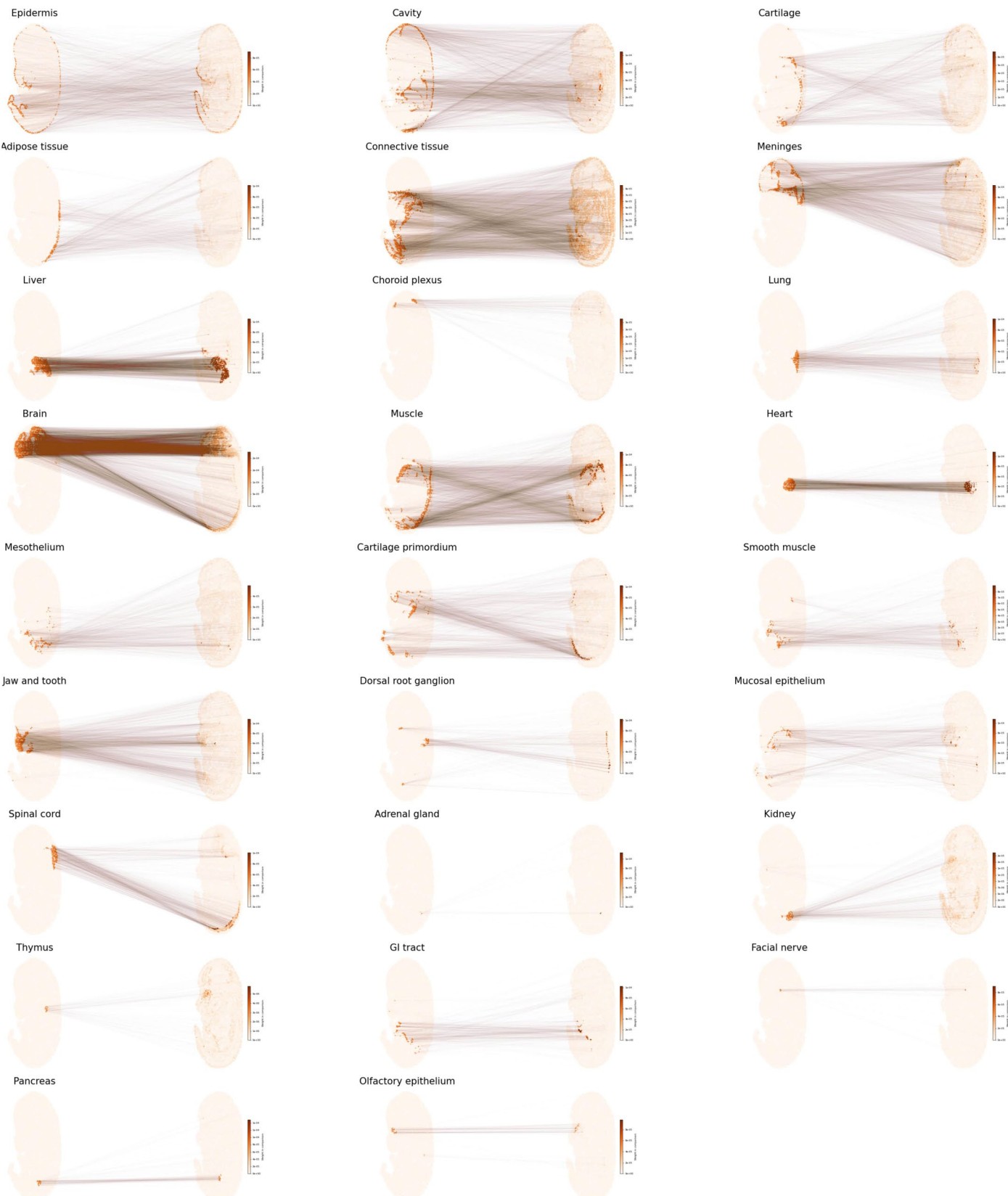

**Extended Data Fig. 6 | Qualitative assessment of PRESCIENT's gene expression predictions in the mouse development dataset.** Visual representation of the linear optimal transport matching between the gene expression predicted by PRESCIENT and the target gene expression measured at the next time point for each cell type present at this time point of the mouse development dataset. For each example, the left slice displays the positions of cells belonging to that cell population and the right slice displays the cells they are matched with at the following time point.

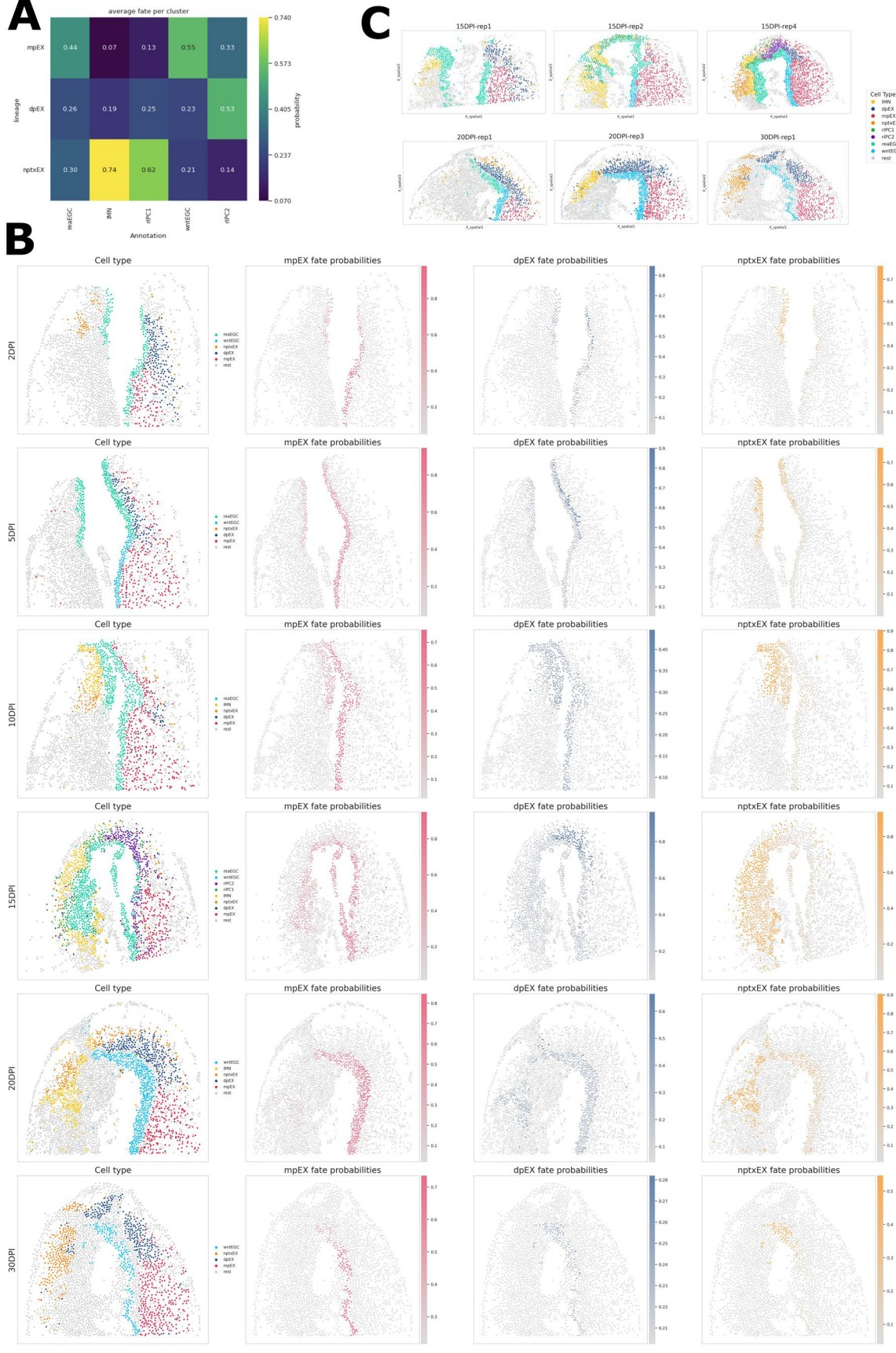

**Extended Data Fig. 7 | See next page for caption.**

**Extended Data Fig. 7 | Predicted fates and replicate overview in axolotl regeneration.** (**A**) Average predicted fate probabilities for each cell type in the axolotl regeneration case study, with rows representing the three possible terminal fates; (**B**) 2-D visualization of the axolotl slices at all time points (one for each row), with colors in the first column corresponding to the studied cell types. In the second to fourth columns, the same slices are represented, with reaEGCs colored by their predicted fate probabilities for mpEX, dpEX and npxTX fates, from left to right; (**C**) All available replicates (which weren't used in the main Figures) for the 15, 20 and 30DPI time points of the axolotl regeneration case study, with cells colored by their cell type annotation.

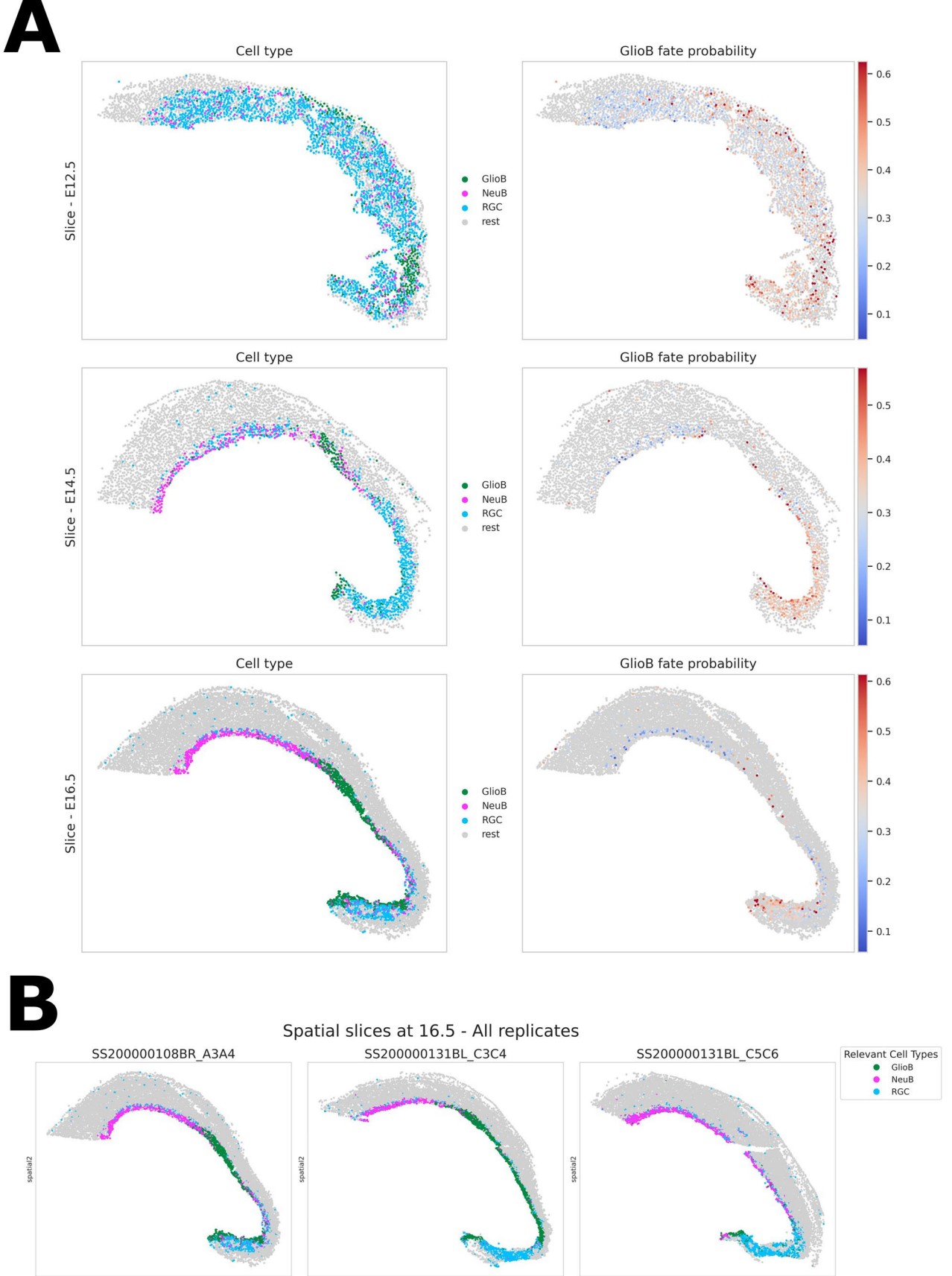

**Extended Data Fig. 8 | GlioB fate prediction and replicate visualization in mouse midbrain development.** (**A**) 2-D visualization of the mouse midbrain slices at all time points (one for each row), with colors in the first column corresponding to the studied cell types. In the second column the same slice is represented and only EGCs are colored by their predicted probability of committing to the GlioB fate; (**B**) 2-D visualization of all available replicate slices of the mouse midbrain at E16.5, with cells colored by their cell type annotation. The first replicate on the left is the one used in the main Fig. 4.

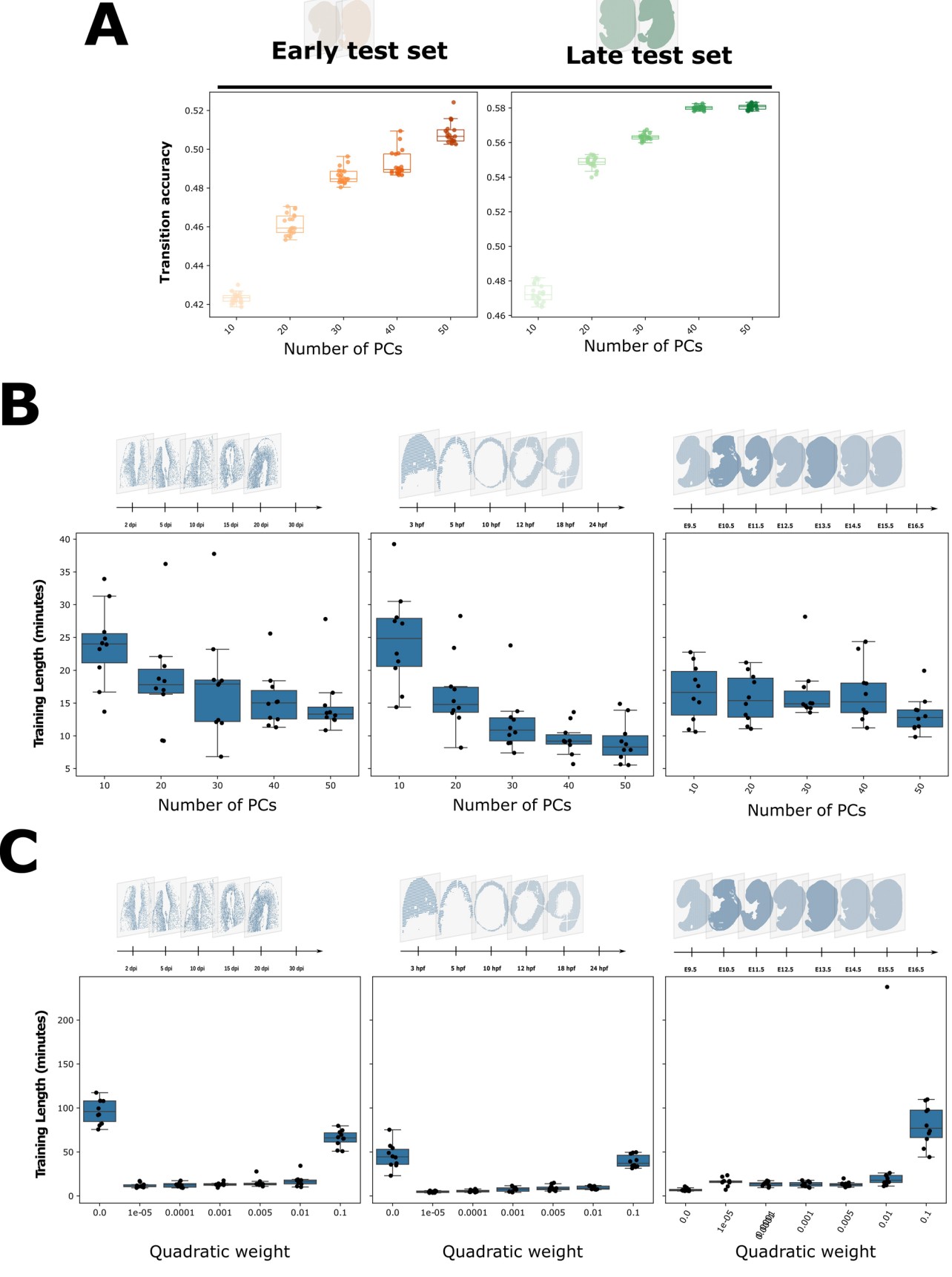

**Extended Data Fig. 9 | See next page for caption.**

**Extended Data Fig. 9 | Effect of input dimensionality and quadratic regularization on accuracy and training time.** In the boxplots, the center line, box limits, and whiskers denote the median, upper and lower quartiles, and 1.5× interquartile range, respectively. All scores are reported for n = 10 initialization seeds. (**A**) Cell type transition accuracy scores in early test set (orange) and late test set (green) of the mouse development dataset for different numbers of principal components used by STORIES for its gene expression input; (**B**) Training duration of STORIES on the training split of (from left to right) the axolotl regeneration, zebrafish development and mouse development datasets. The computational runtimes are compared across the same choices of number of principal components as in the first panel; (**C**) Training duration of STORIES on the training split of (from left to right) the axolotl regeneration, zebrafish development and mouse development datasets. Computational runtimes are com- pared across seven values of quadratic weight α parameter, including the linear method (α = 0, light orange/green).

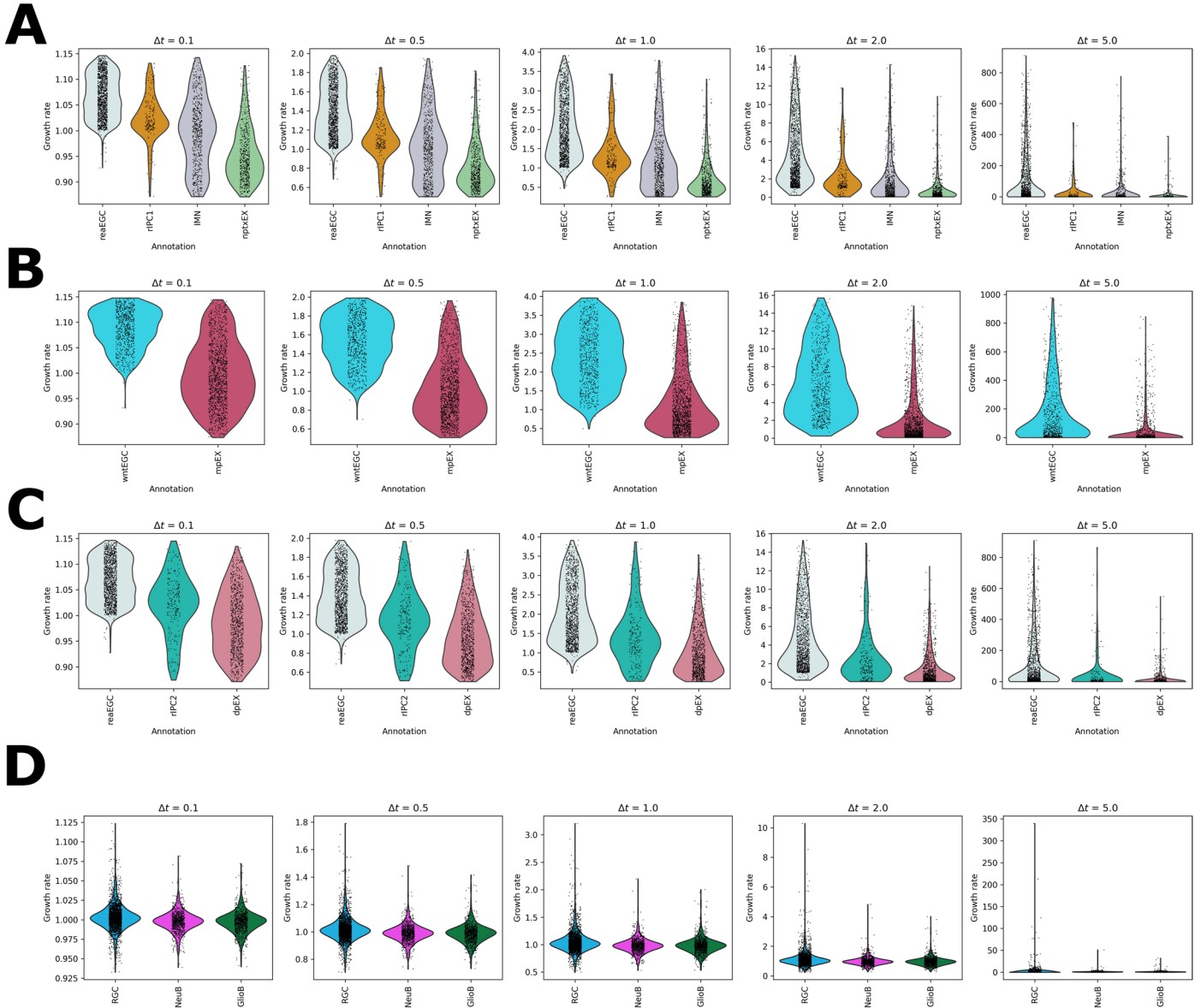

**Extended Data Fig. 10 | Estimated growth rates across trajectories and ∆t parameter values. (A)** In the axolotl regeneration case study, we report for different choices of the ∆t parameter the estimated growth rates for all cell populations taking part in the reaEGC-rIPC1-IMN-nptxEX trajectory. **(B)** In the axolotl regeneration case study, we report for different choices of the ∆t parameter the estimated growth rates for all cell populations taking part in the wntEGC-mpEX trajectory. **(C)** In the axolotl regeneration case study, we report for different choices of the ∆t parameter the estimated growth rates for all cell populations taking part in the reaEGC-rIPC2-dpEX trajectory. **(D)** In the mouse midbrain case study, we report for different choices of the ∆t parameter the estimated growth rates for all three studied cell populations.

# Reporting Summary

## Statistics

For all statistical analyses, confirm that the following items are present in the figure legend, table legend, main text, or Methods section.

| n/a | Confirmed | |
|---|---|---|
| ☐ | ☒ | The exact sample size (*n*) for each experimental group/condition, given as a discrete number and unit of measurement |
| ☐ | ☒ | A statement on whether measurements were taken from distinct samples or whether the same sample was measured repeatedly |
| ☐ | ☒ | The statistical test(s) used AND whether they are one- or two-sided<br>*Only common tests should be described solely by name; describe more complex techniques in the Methods section.* |
| ☒ | ☐ | A description of all covariates tested |
| ☒ | ☐ | A description of any assumptions or corrections, such as tests of normality and adjustment for multiple comparisons |
| ☐ | ☒ | A full description of the statistical parameters including central tendency (e.g. means) or other basic estimates (e.g. regression coefficient) AND variation (e.g. standard deviation) or associated estimates of uncertainty (e.g. confidence intervals) |
| ☐ | ☒ | For null hypothesis testing, the test statistic (e.g. *F*, *t*, *r*) with confidence intervals, effect sizes, degrees of freedom and *P* value noted<br>*Give P values as exact values whenever suitable.* |
| ☒ | ☐ | For Bayesian analysis, information on the choice of priors and Markov chain Monte Carlo settings |
| ☒ | ☐ | For hierarchical and complex designs, identification of the appropriate level for tests and full reporting of outcomes |
| ☒ | ☐ | Estimates of effect sizes (e.g. Cohen's *d*, Pearson's *r*), indicating how they were calculated |

*Our web collection on statistics for biologists contains articles on many of the points above.*

## Software and code

Policy information about availability of computer code

| Data collection | No software was used for data collection |
|---|---|
| Data analysis | Python v3.10<br><br>Custom package: stories v0.1.0<br>Package repository: https://github.com/cantinilab/stories<br>Reproducibility repository: https://github.com/cantinilab/stories_reproducibility<br><br>Other packages:<br>- scanpy v1.10.1<br>- anndata v0.10.7<br>- jax v0.4.25<br>- orbax-checkpoint v0.5.9<br>- optax v0.2.2<br>- flax v0.8.2<br>- jaxopt v0.8.3<br>- ott-jax v0.4.5<br>- tqdm v4.66.2<br>- numpy v1.26.4<br>- scikit-learn v1.5.1<br>- magic-impute v3.0.0 |

```
- seaborn v0.13.2
- cellrank v2.0.4
```

For manuscripts utilizing custom algorithms or software that are central to the research but not yet described in published literature, software must be made available to editors and reviewers. We strongly encourage code deposition in a community repository (e.g. GitHub). See the Nature Portfolio guidelines for submitting code & software for further information.

## Data

Policy information about availability of data

All manuscripts must include a data availability statement. This statement should provide the following information, where applicable:

- Accession codes, unique identifiers, or web links for publicly available datasets
- A description of any restrictions on data availability
- For clinical datasets or third party data, please ensure that the statement adheres to our policy

- We retrieved the mouse Stereo-seq atlas from Chen et al., available at https://db.cngb.org/stomics/mosta/.
- We retrieved the zebrafish Stereo-seq atlas from Liu et al., available at https://db.cngb.org/stomics/zesta/.
- We retrieved the axolotl Stereo-seq atlas from Wei et al., available at https://db.cngb.org/stomics/artista/.
- We retrieved mouse proliferation genes from Tirosh et al. (see Supplementary Text 1) and apoptosis genes from gsea-msigdb's HALLMARK_P53_PATHWAY (see Supplementary Text 2), available at https://www.gsea-msigdb.org/gsea/msigdb/human/geneset/HALLMARK_P53_PATHWAY.html.
- We retrieved an NSC axolotl gene set from Wei et al. (see Supplementary Text 3) and a human apoptosis gene set from gsea-msigdb's HALLMARK_APOPTOSIS (see Supplementary Text 4), available at https://www.gsea-msigdb.org/gsea/msigdb/human/geneset/HALLMARK_APOPTOSIS.html.

## Research involving human participants, their data, or biological material

Policy information about studies with human participants or human data. See also policy information about sex, gender (identity/presentation), and sexual orientation and race, ethnicity and racism.

| | |
|---|---|
| Reporting on sex and gender | Not applicable. We apply our computational method to public data. No human data was used. |
| Reporting on race, ethnicity, or other socially relevant groupings | Not applicable. We apply our computational method to public data. No human data was used. |
| Population characteristics | Not applicable. We apply our computational method to public data. No human data was used. |
| Recruitment | Not applicable. We apply our computational method to public data. No human data was used. |
| Ethics oversight | Not applicable. We apply our computational method to public data. No human data was used. |

Note that full information on the approval of the study protocol must also be provided in the manuscript.

# Field-specific reporting

Please select the one below that is the best fit for your research. If you are not sure, read the appropriate sections before making your selection.

☒ Life sciences    ☐ Behavioural & social sciences    ☐ Ecological, evolutionary & environmental sciences

For a reference copy of the document with all sections, see nature.com/documents/nr-reporting-summary-flat.pdf

# Life sciences study design

All studies must disclose on these points even when the disclosure is negative.

| | |
|---|---|
| Sample size | No "life sciences" experiments were done in this study. The only samples which were used were the random seeds used to initialize the computational methods in the benchmark.<br>We did not perform a formal statistical power analysis to predetermine the number of random seeds. Instead, we selected 10 random seeds in line with common practice in machine learning benchmarks, where multiple runs are used to account for variance due to stochastic elements such as parameter initialization, data splits, and optimization dynamics. Running each method with 10 independent seeds allowed us to estimate performance variability and ensure robust and reproducible comparisons across methods. |
| Data exclusions | For each dataset, a single slice for each time point was analyzed. The choice of slice is based on the original publications (see Methods). The analysis represented in Figures 3 and 4 only considers cell types of interest, as in the original publications (see Methods). |
| Replication | All experiments were run for 10 random seeds, represented in boxplots in Figure 2, Extended Data Figures 2 & 9. |
| Randomization | Train/validation splits, parameter initialization, and minibatch sampling are controlled by a randomly chosen seed. All experiments were run for the same 10 random seeds. |
| Blinding | Not applicable. We apply our computational method to public data. Our study doesn't involve group allocation that requires blinding. |

# Reporting for specific materials, systems and methods

We require information from authors about some types of materials, experimental systems and methods used in many studies. Here, indicate whether each material, system or method listed is relevant to your study. If you are not sure if a list item applies to your research, read the appropriate section before selecting a response.

## Materials & experimental systems

| n/a | Involved in the study |
|-----|-----------------------|
| ☒ | Antibodies |
| ☒ | Eukaryotic cell lines |
| ☒ | Palaeontology and archaeology |
| ☒ | Animals and other organisms |
| ☒ | Clinical data |
| ☒ | Dual use research of concern |
| ☒ | Plants |

## Methods

| n/a | Involved in the study |
|-----|-----------------------|
| ☒ | ChIP-seq |
| ☒ | Flow cytometry |
| ☒ | MRI-based neuroimaging |

## Plants

| | |
|---|---|
| Seed stocks | *Report on the source of all seed stocks or other plant material used. If applicable, state the seed stock centre and catalogue number. If plant specimens were collected from the field, describe the collection location, date and sampling procedures.* |
| Novel plant genotypes | *Describe the methods by which all novel plant genotypes were produced. This includes those generated by transgenic approaches, gene editing, chemical/radiation-based mutagenesis and hybridization. For transgenic lines, describe the transformation method, the number of independent lines analyzed and the generation upon which experiments were performed. For gene-edited lines, describe the editor used, the endogenous sequence targeted for editing, the targeting guide RNA sequence (if applicable) and how the editor was applied.* |
| Authentication | *Describe any authentication procedures for each seed stock used or novel genotype generated. Describe any experiments used to assess the effect of a mutation and, where applicable, how potential secondary effects (e.g. second site T-DNA insertions, mosiacism, off-target gene editing) were examined.* |

