## [Peer Review File · Nature Methods]

STORIES: learning cell fate landscapes from spatial transcriptomics using optimal transport

Corresponding Author: Dr Laura Cantini

Version 0:

Decision Letter:

20th Sep 2024

Dear Dr Cantini,

Your Article, "Learning cell fate landscapes from spatial transcriptomics using Fused Gromov-Wasserstein", has now been seen by 2 reviewers. As you will see from their comments below, although the reviewers find your work of potential interest, they have raised a number of very important concerns. We are interested in the possibility of publishing your paper in Nature Methods, but would like to consider your response to these concerns before we reach a final decision on publication.

We therefore invite you to extensively revise your manuscript to fully address all these concerns. Among other revisions, we think it is important to show strong performance advance compared to other methods and application results with strong biological significance/interest. We will need to evaluate the revisions before deciding on whether it would be suitable to send the revised paper back for re-review.

Link Redacted

We hope to receive your revised paper within 4 months. If you cannot send it within this time, please let us know. In this event, we will still be happy to reconsider your paper at a later date so long as nothing similar has been accepted for publication at Nature Methods or published elsewhere.

OPEN SCIENCE REQUIREMENTS

REPORTING SUMMARY AND EDITORIAL POLICY CHECKLISTS

DATA AVAILABILITY

All novel DNA and RNA sequencing data, protein sequences, genetic polymorphisms, linked genotype and phenotype data, gene expression data, macromolecular structures, and proteomics data must be deposited in a publicly accessible database, and accession codes and associated hyperlinks must be provided in the "Data Availability" section.

CODE AVAILABILITY

Please include a "Code Availability" subsection in the Online Methods which details how your custom code is made available. Only in rare cases (where code is not central to the main conclusions of the paper) is the statement "available upon request" allowed (and reasons should be specified).

MATERIALS AVAILABILITY

ORCID

Nature Methods is committed to improving transparency in authorship. As part of our efforts in this direction, we are now

requesting that all authors identified as 'corresponding author' on published papers create and link their Open Researcher and Contributor Identifier (ORCID) with their account on the Manuscript Tracking System (MTS), prior to acceptance. This applies to primary research papers only. ORCID helps the scientific community achieve unambiguous attribution of all scholarly contributions. You can create and link your ORCID from the home page of the MTS by clicking on 'Modify my Springer Nature account'. For more information please visit www.springernature.com/orcid.

Sincerely,

Lin Tang, PhD
Senior Editor
Nature Methods

Reviewers' Comments:

Reviewer #1 (Remarks to the Author):

This paper presents a tool to construct a Waddington landscape on both gene expression and spatial locations by optimizing a neural network on a temporal sequence of spatial transcriptomics data. Fused-Gromov-Wasserstein distance is used as the loss function where the Wasserstein term quantifies gene expression differences and the GW term quantifies spatial arrangement conservation. While I find the idea interesting, I have several major concerns. It is not clear what is the practical benefit compared to existing approaches that are likely computationally more efficient (aligning first and then OT). Also, currently presented results only show marginal difference on inferred trajectories and the advantage of including the spatial component is not clear. Furthermore, it would be very helpful to show spatial interpretations of the reconstructed spatio-temporal trajectories, in addition to the current post-analysis on only the temporal aspect.

1. In the second paragraph on page 3, it was mentioned that "existing methods for trajectory inference based on Wasserstein gradient flows are not equipped to deal with spatially resolved omics data." Does the difficulty come from the extra spatial information and unaligned original data? I wonder what are the benefits of the proposed approach in practice compared to first aligning the slices and perform a standard dynamical OT (stVCR). When using standard dynamical OT, the potential can be conveniently obtained. While other methods might not have potential implemented in their software, it would still be helpful to compare with stVCR and other related methods on the inferred trajectories.

2. In the second section in Results, STORIES was compared to other linear OT approaches. A special case of STORIES ($\alpha=0$) was used to represent three other methods. While they may share the same or very similar OT model in this special case, the implementation, parameterization, and optimization may be different. For example, the energy J in (Bunne et al. 2022) may be parameterized with a different neural network architecture. The same issue applies to the other two existing methods mentioned. For formal method comparison, detailed setups or software packages (if available) of methods been compared to should be used, rather than simply representing other methods with STORIES with $\alpha=0$.

3. The results presented in Figure 2B are less convincing about the improvement on gene expression prediction due to incorporating spatial information, where only one of six cases show improvements. I understand that methodologically, this could happen due to how the model is setup. However, this contradicts with one of the overarching biological goals of improving trajectory prediction/interpolation by incorporating the additional spatial information.

4. This is related to the previous point. The differences in the scores (Figure 2b) are quite small (whether improvement or not). It would be helpful to show examples where STORIES corrects qualitatively "wrong" trajectories identified by existing approaches.

5. In the trajectory plot in Figure 3B, it is hard to tell apart "reaEGC-rIPC2-dpEX" vs "wntEGC-mpEX". Could you add a figure panel of the cell-cell transition matrix summarized into cell types?

6. I believe one of the major strength of incorporating spatial information in trajectory analysis is to decipher how spatial environment impacts with cell fate decisions. However, there is little discussion on spatial interpretation of the predicted trajectories, other than some quantitative and relatively marginal improvements on the scores in Figure 2. Adding analysis of this kind could provide new analysis utilities. For example, in the second case study, analyzing whether and how the branching decision is related to differences in spatial environment will be very helpful.

7. While the growth rate estimation has been used in several existing works, it is not gold standard yet. Robustness of the results with respect to different parameter choices in the growth rate empirical function should be demonstrated.

8. Could you clarify computational costs? FGW is computationally costly and I guess incorporating it as a loss term in deep learning further increases the cost. Is the tool feasible for biologists with normal workstation to analyze a sequence of Stereo-seq data each containing thousands of cells? Adding similar plots as Supplementary Figure 1 but with non-zero α will be helpful.

9. In evaluation, instead of using Wasserstein distance for gene expression score and GW distance for spatial score, why using FGW with a single α parameter to determine both scores? IMHO, the earlier approach is more natural.

10. While modeling cell state dynamics with Waddington landscape is widely accepted, could you clarify why the spatial component can also be modeled as an energy potential (i.e., curl-free)? It would be helpful to highlight the biological interpretation of this assumption.

Reviewer #2 (Remarks to the Author):

Summary:

The paper presents SpatioTemporal Omics eneRgIES (STORIES), a novel method designed to infer cell fate trajectories from spatiotemporal transcriptomics data using Fused Gromov-Wasserstein (FGW). This method models cell differentiation by learning a potential landscape that integrates spatial and gene expression data, providing predictions for future cellular states and gene trends. The method is benchmarked on large-scale spatiotemporal datasets, such as axolotl neural regeneration and mouse gliogenesis. The method demonstrates superior spatial coherence relative to its linear non-spatial counterpart and shows it is capable of generating biologically relevant insights.

Strengths:

The method improves upon existing OT methods by integrating both spatial and gene expression data. The method applies an FGW cost which allows the model to incorporate spatial information up to isometries.

Some nice applications are given, including an analysis of trajectories in mouse dorsal midbrain development. A few interesting genes are identified for further investigation in gliogenesis.

The comparison to the linear method clearly highlights the usefulness of incorporating spatial coordinates, showing that STORIES's inclusion of spatial information enhances spatial coherence and predictive accuracy across multiple datasets. The results emphasize the value of integrating spatial data in the loss function.

The method is provided as an open-source Python package, enabling integration with existing single-cell analysis tools, making it accessible for broader use.

Limitations/Weaknesses/Questions/Comments:

As the authors note in the discussion, the potential function does not incorporate spatial data directly and generate predictions in spatial coordinates, so it does not model the physical migration of cells. Also, the use of a potential-based model might oversimplify complex biological processes, such as cell-cell communication or oscillations within cell states. Although not the primary goal of the article, addressing these aspects would significantly improve the impact of the manuscript.

The method is evaluated on 50 PCA components from 10k highly variable genes. How is training time and performance impacted when using all genes instead of the 10k highly variable ones? How is the training time and performance impacted when using more PCA components than 50?

Y-axis labels for the scatter plots would help readability

Version 1:

Decision Letter:

Our ref: NMETH-A57311A

29th Apr 2025

Dear Dr. Cantini,

We very much apologize for the longer peer review process than we would hope for your revised manuscript "Learning cell fate landscapes from spatial transcriptomics using Fused Gromov-Wasserstein" (NMETH-A57311A). Thank you for revising and submitting it, which has now been seen by the original referees whose comments are below. The reviewers find that the paper has improved in revision, and therefore we'll be happy in principle to publish it in Nature Methods, pending minor revisions to satisfy the referees' final requests and to comply with our editorial and formatting guidelines.

TRANSPARENT PEER REVIEW

ORCID

Sincerely,

Lin Tang, PhD
Senior Editor
Nature Methods

Reviewer #1 (Remarks to the Author):

The authors have addressed all my concerns. I believe the manuscript has been significantly improved and ready for publication.

Reviewer #2 (Remarks to the Author):

Thank you for providing the additional discussion section on potential extensions and their current obstacles, which add color to the contributions of the current manuscript. The authors have clarified my understanding of the role of the FGW loss in the context of incorporating the spatial information in STORIES.

- One additional point of clarification: in the methods section on "Learning the Potential", is the notation used for the different sets of spatial coordinates, "r" and "s", meant for a general formulation of the quadratic term in the FGW formulation? And am I correct in that the r and s are the same when optimizing f_{θ} as shown on line 638, since STORIES does not predict explicitly spatial coordinates? Perhaps a sentence just clarifying what r and s are in practice for STORIES would be useful.

The additional metrics provided in response to the other reviewer have enhanced the story in my view. While I agree with the other reviewer that the improvement in the approach is not groundbreaking, it is still a nontrivial improvement, and I believe the method is well-justified and provides a step in the right direction regarding the difficult task of modeling cell fates. I would support acceptance for this manuscript in its current state.

Version 2:

Decision Letter:

2nd Sep 2025

Dear Dr Cantini,

Thank you very much for sending us the updated version of your Article, "STORIES: learning cell fate landscapes from spatial transcriptomics using optimal transport". I am pleased to inform you that this paper has now been accepted for publication in Nature Methods. The received and accepted dates will be 26th Jul 2024 and 2nd Sep 2025. This note is intended to let you know what to expect from us over the next month or so, and to let you know where to address any further questions.

Over the next few weeks, your paper will be copyedited to ensure that it conforms to Nature Methods style. Once your paper is typeset, you will receive an email with a link to choose the appropriate publishing options for your paper and our Author Services team will be in touch regarding any additional information that may be required. It is extremely important that you let us know now whether you will be difficult to contact over the next month. If this is the case, we ask that you send us the contact information (email, phone and fax) of someone who will be able to check the proofs and deal with any last-minute problems.

After the grant of rights is completed, you will receive a link to your electronic proof via email with a request to make any corrections within 48 hours. If, when you receive your proof, you cannot meet this deadline, please inform us at

rjsproduction@springernature.com immediately.

Authors may need to take specific actions to achieve compliance with funder and institutional open access mandates.

If your research is supported by a funder that requires immediate open access (e.g. according to [Plan S principles](https://www.springernature.com/gp/open-science/plan-s-compliance) or the [NIH public access policy](https://www.springernature.com/gp/open-science/us-federal-agency-compliance)) then you should select the gold OA route, and we will direct you to the compliant route where possible. Because authors warrant under our subscription licensing terms that they haven't committed to licensing any version of their article under a licence inconsistent with the terms of our agreement – including the applicable embargo period – publication under the subscription model isn't suitable for authors whose funders require no embargo.

Please feel free to contact me if you have questions about any of these points. Thank you very much for publishing your paper at Nature Methods!

Best regards,

Lin Tang, PhD
Senior Editor
Nature Methods

** Visit the Springer Nature Editorial and Publishing website at http://editorial-jobs.springernature.com?utm_source=ejP_NMeth_email&utm_medium=ejP_NMeth_email&utm_campaign=ejp_Nmeth for more information about our career opportunities. If you have any questions please click [here](mailto:editorial.publishing.jobs@springernature.com).

Reviewers' Comments:

Reviewer #1 (Remarks to the Author):

This paper presents a tool to construct a Waddington landscape on both gene expression and spatial locations by optimizing a neural network on a temporal sequence of spatial transcriptomics data. Fused-Gromov-Wasserstein distance is used as the loss function where the Wasserstein term quantifies gene expression differences and the GW term quantifies spatial arrangement conservation. While I find the idea interesting, I have several major concerns. It is not clear what is the practical benefit compared to existing approaches that are likely computationally more efficient (aligning first and then OT). Also, currently presented results only show marginal difference on inferred trajectories and the advantage of including the spatial component is not clear. Furthermore, it would be very helpful to show spatial interpretations of the reconstructed spatio-temporal trajectories, in addition to the current post-analysis on only the temporal aspect.

1. In the second paragraph on page 3, it was mentioned that “existing methods for trajectory inference based on Wasserstein gradient flows are not equipped to deal with spatially resolved omics data.” Does the difficulty come from the extra spatial information and unaligned original data? I wonder what are the benefits of the proposed approach in practice compared to first aligning the slices and perform a standard dynamical OT (stVCR). When using standard dynamical OT, the potential can be conveniently obtained. While other methods might not have potential implemented in their software, it would still be helpful to compare with stVCR and other related methods on the inferred trajectories.

Reply: The existing methods do not model the space. While technically they could be applied to the concatenation of the gene expression measurements and spatial coordinates, this would be problematic as spatial coordinates are incomparable across timepoints. Indeed, when handling spatial coordinates of unaligned slices, one must account for their translational and rotational invariances. Rigidly aligning the slices, as is done in stVCR aims at fixing this issue. However, this method doesn't account for the dilatations and other deformations that can occur in developing organisms. For example, in our zebrafish dataset the embryos undergo a total morphological transformation during their development which can't be modeled with a rigid alignment as in stVCR. More generally, obtaining a representation for spatial coordinates which is coherent and comparable across all timepoints is a challenging issue. Since a Procrustes alignment of the slices is not sufficient to solve it, stVCR learns a velocity function for the space coordinates which requires the time point of the samples as an additional input. This enables the spatial velocity function to learn internally for each time point a representation correcting the discrepancies which remain after the rigid alignment of the slices. This strategy might impair stVCR's generalization performance on unseen slices of later timepoints. On the other hand, Gromov Wasserstein (GW) is properly suited to compare slices across timepoints since its transport plan is invariant to rotating, translating and rescaling of the spatial coordinates. On top of that, GW allows us to implicitly guide our learned potential to depend on the space information without having to use a representation of the space as input of the potential. We can

therefore learn a differentiation potential shared across all time points and model a general dynamic less prone to overfitting.

It would be useful to prove this empirically by comparing the approaches but unfortunately there's no code available to train stVCR. The authors released notebooks where they load checkpoints of their model trained on specific datasets but it's still not possible to train and apply the method to a new dataset. There exist no other method outside of stVCR and Stories, to the best of our knowledge, which attempt to rely on spatial information in order to perform trajectory inference apart from SpaTrack which learns velocities based on linear OT taking into account both space and gene expression. However SpaTrack doesn't learn velocity as a parameterized function and therefore it cannot be used to extrapolate to new timepoints in order to predict future gene expressions.

We now clarify both in the introduction and in the discussion of the paper the challenges added by the use of the space in trajectory inference and explain why we can't use stVCR in the benchmark.

2. In the second section in Results, STORIES was compared to other linear OT approaches. A special case of STORIES ($\alpha=0$) was used to represent three other methods. While they may share the same or very similar OT model in this special case, the implementation, parameterization, and optimization may be different. For example, the energy J in (Bunne et al. 2022) may be parameterized with a different neural network architecture. The same issue applies to the other two existing methods mentioned. For formal method comparison, detailed setups or software packages (if available) of methods been compared to should be used, rather than simply representing other methods with STORIES with $\alpha=0$.

Reply: The choice of using STORIES ($\alpha=0$) as the reference with which to compare our performances comes from three main reasons:

1. We wanted to assess strictly the added value of the space with respect to the purely linear model. To do so, we aimed for a fair comparison using the same formalization for the baseline ($\alpha=0$) and the quadratic model ($\alpha > 0$). Implementations of the linear OT model mainly differ in three aspects: the number of intermediary predictions between two-time points (PRESCIENT¹ and Hashimoto et al.² use several while JKOnet³ only uses one), the use of teacher forcing (JKOnet uses it while PRESCIENT and Hashimoto et al. don't) and the discretization scheme (PRESCIENT and Hashimoto et al. use an explicit forward Euler discretization scheme while JKOnet propose an implicit backward Euler scheme). We compared all the possible implementations of the linear OT model in Supplementary Figure 1 and selected the best-performing one for the baseline. This means that our baseline (STORIES with $\alpha=0$) combines the best elements of existing linear OT methods. We then used the same implementation as the backbone of the quadratic model.
2. Most existing linear OT methods don't have an available code implementation for our task. TrajectoryNet⁴ which is based on dynamicalOT can only perform

interpolation and doesn't implement next time point prediction. JKOnet provides reproducibility scripts for their paper but the method can't be applied to new single-cell data as there is the need to reimplement several parts of the training procedure such as a single-cell dataloader. Similarly, Hashimoto et al. only provide a reproducibility repository without instructions on how to apply the method to new data and which parameters to choose. Moreover, PRESCIENT was created by co-authors of Hashimoto et al. based on the same principles and has a well-documented codebase. PRESCIENT is thus the only state-of-the-art method that could be compared to STORIES on our benchmark datasets. We now included PRESCIENT in the benchmark and significantly outperformed it. Importantly, PRESCIENT's default parameter handling the scaling of the noise was not adapted to our applications. We therefore tuned it to evaluate PRESCIENT to its full potential. STORIES outperforms PRESCIENT even after this parameter tuning.

We now integrated the discussion regarding coding implementations of the linear OT-based models in the end of the first section of the results and updated the benchmark to include PRESCIENT.

3. The results presented in Figure 2B are less convincing about the improvement on gene expression prediction due to incorporating spatial information, where only one of six cases show improvements. I understand that methodologically, this could happen due to how the model is setup. However, this contradicts with one of the overarching biological goals of improving trajectory prediction/interpolation by incorporating the additional spatial information.

Reply: The main goal of this Figure was to display the tradeoff between spatial coherence and gene expression error. For the best performing values of α (0,001-0,005) we found that in 4 out of 6 examples (axolotl early test, zebrafish late test, mouse early and late test) we could achieve both lower spatial and gene expression errors, when looking at the median.

However, in his/her 9th remark, the Reviewer raised the question of using the Wasserstein distance to assess the quality of gene expression predictions. Since it directly evaluates the downstream performance of the model and is now a classical metric in trajectory inference literature^{1,3-6}, we included it in the benchmark.

With this metric, STORIES leads to significant improvements with respect to both the state-of-the-art method PRESCIENT and STORIES ($\alpha=0$), see Figure 2B. To further demonstrate the added value brought by STORIES by incorporating spatial information, we evaluated cell type transition accuracy in the mouse dataset.

This was not possible for all datasets because there are discrepancies between the set of cell populations present at different time points. Indeed, during the development of living organisms some cell types appear, disappear or transform into new cell types. In the mouse dataset the authors of MOSCOT⁷ performed an extensive literature search in order to annotate cell type transitions which were biologically valid. We used this ground truth to evaluate the cell type transitions predicted by STORIES and PRESCIENT. For this evaluation, we used the gene expression predicted by each

method at time $t+1$, then computed a transport plan between this predicted gene expression and the ground truth measurements at $t+1$. This transport plan was then used as a similarity matrix on which we applied k-nn classification. This resulted in cell type predictions which we compared with the ground truth. Importantly, those cell type predictions are only based on gene expression predictions and didn't use spatial coordinates as an input, neither in the potential nor in the transport plan. The spatial information was only leveraged by STORIES during the training, and not at prediction time. STORIES also significantly outperforms PRESCIENT based on the cell type transition accuracy: 50% vs. 41% in the early test set and 58% vs. 46% in the late test set, see Figure 2C. This improvement in performances is due to STORIES' ability to guide the learning of the gene expression potential with a space-informed loss.

We now update Figure 2 reporting the new scores using 50PCs as input (as a result of Reviewer 2's remark) for the benchmark evaluation. Additional plots regarding the quantitative evaluation of STORIES are now displayed in Supplementary Figure 2. The discussion above is integrated in the modified manuscript (second section of results) and used scores are explained in the online methods.

4. This is related to the previous point. The differences in the scores (Figure 2b) are quite small (whether improvement or not). It would be helpful to show examples where STORIES corrects qualitatively "wrong" trajectories identified by existing approaches.

Reply: Based on the arguments mentioned in our reply to point 3, we really believe that the current evaluation with the Wasserstein distance and cell type transition accuracy clearly shows an added value on the quantitative side.

As requested by the Reviewer we still added also qualitative examples in Figure 2D. Supplementary Figures 5-10 report the comparison between all predictions made by PRESCIENT and STORIES in zebrafish (between timepoints 12HPF and 18HPF), mouse (between time points E14.5 and E15.5) and axolotl (between time points 15DPI and 20DPI).

In particular, regarding wrong predictions made by existing approaches (see Figure 2C):

- In zebrafish development when transitioning from 12HPF and 18HPF PRESCIENT wrongly predicts Adaxial cells to be distributed all over the embryo. On the contrary, STORIES correctly matches Adaxial cells with cells close to the notochord. Indeed, adaxial cells are known to reside next to the notochord (<https://www.sciencedirect.com/science/article/pii/S0361923000003828> <https://journals.biologists.com/dev/article/122/11/3371/38862/Identification-of-separate-slow-and-fast-muscle>). In addition, Adaxial cells are known to differentiate into slow muscle cells (<https://www.sciencedirect.com/science/article/pii/S0361923000003828>) and indeed based on the annotation of 24HPF (see Supplementary Figure 4), slow muscle cells are localized in the same area where STORIES predicts Adaxial cells to evolve.

- In zebrafish development when transitioning from 12HPF and 18HPF PRESCIENT wrongly predicts somite cells to match only with the tail area missing to identify most of the somite cells annotated in the 18HPF zebrafish embryo. On the opposite, STORIES correctly matches somite cells from 12HPF with all somite cells from 18HPF.

In addition, as shown in Supplementary Figures 7 and 8 also in axolotl transition from 15DPI to 20DPI, we can observe a stronger mapping for *wntEGCs* and *RrIPC1* cells in STORIES with respect to PRESCIENT. In the same way, in mouse transition from E14.5 to E15.5, we can observe a stronger mapping for “Choroid plexus” and “jaw and tooth” in STORIES with respect to PRESCIENT (Supplementary Figures 9 and 10).

All these qualitative examples show that STORIES corrects qualitatively “wrong” trajectories identified by existing approaches.

We now updated Figure 2 to report such qualitative examples, added Supplementary Figures 5-10 to show the overall qualitative behaviour in the three model systems and discussed these results in section 2.

5. In the trajectory plot in Figure 3B, it is hard to tell apart “*reaEGC-rIPC2-dpEX*” vs “*wntEGC-mpEX*”. Could you add a figure panel of the cell-cell transition matrix summarized into cell types?

Reply: Indeed it is hard to tell apart “*reaEGC-rIPC2-dpEX*” vs “*wntEGC-mpEX*”, because biologically these are not independent trajectories. For example, as shown also in Figure S14 panel J of their original paper⁸ *reaEGCs* can become *mpEX*. In addition, in the results Wei X. et al. show that *reaEGCs* restore *nptxEXs* after injury.

To clarify the transitions between cell populations, we used CellRank⁹ to compute fate probabilities based on the velocity derived from our learn potentials. For each of the three terminal states, we then aggregated those cell-level probabilities at the level of cell types. As shown in Supplementary Figure 11, *reaEGCs* have comparable chances to become *mpEX*, *dpEX* and *nptxEX*.

We now modified the text of section 3 to specify that the three mentioned trajectories are not comprehensive of all possible cell fates of *reaEGCs* and that another visualization of cell type transitions is now available in Supplementary Figure 11.

6. I believe one of the major strength of incorporating spatial information in trajectory analysis is to decipher how spatial environment impacts with cell fate decisions. However, there is little discussion on spatial interpretation of the predicted trajectories, other than some quantitative and relatively marginal improvements on the scores in Figure 2. Adding analysis of this kind could provide new analysis utilities. For example, in the second case study, analyzing whether and how the branching decision is related to differences in spatial environment will be very helpful.

Reply: We agree with the Reviewer regarding the importance of this point. We now added two new panels in Figures 3 and 4 showing possible relevant examples of how the environment influences cell fate decisions and discussed the results in subsections 3 and 4 of results.

In Figure 3C-D for axolotl, focusing on time point 15DPI (see Supplementary Figure 13 for other time points), we show that the reaEGCs have a different cell fate depending on their spatial location in the slice. ReaEGCs on the right of the injury tend to commit more towards mpEX, while ReaEGCs on the left of the injury tend to commit more towards nptxEX. This observation seems to be in accordance with the spatial location of the terminal states in the slice. In addition, this same spatial organization of the terminal states can be observed in the additional two replicates available (see Supplementary Figure 13), supporting its biological relevance.

In Figure 4C-D for mouse midbrain, focusing on time point E16.5 (see Supplementary Figure 14 for other time points), the differentiation of Radial Glial Cells (RGCs) into either neuroblasts (NeuB) or glioblasts (GlioB) seems to be influenced by their spatial location. RGCs on the rostral part tend to commit to NeuB, RGCs on the extreme side of the caudal part tend to commit to GlioB. Finally, RGCs on the central part tend to be organized into clusters of cells that either commit to NeuB or to GlioB. Importantly, these conclusions are supported by the agreement between the spatial position of the already differentiated cells and our predicted fate probabilities for the RGCs. These same observations apply for time point E14.5 (see Supplementary Figure 15).

All the analysis on the fate probabilities was performed using CellRank⁹ on the velocity derived from our learned potential (see Methods).

Altogether these results prove that the spatial environment impacts cell fate decisions in both the two discussed test cases. As a consequence, STORIES, taking into account the space in trajectory inference, is better suited for cell fate inference.

7. While the growth rate estimation has been used in several existing works, it is not gold standard yet. Robustness of the results with respect to different parameter choices in the growth rate empirical function should be demonstrated.

Reply: We want first to clarify that the growth rate estimation has not been used in the benchmarking of STORIES (section 2 of the results). This choice was done to have a comparison with the state-of-the-art that is only focused on the quality of their underlying model. Indeed, other state-of-the-art methods sometimes use alternative ways to model the growth rate^{1,4} or do not consider it at all⁶. In addition, the growth rate estimation requires knowing a gene signature that is well established for some organisms, but might be missing for less studied ones.

Growth rate estimation has only been added in the two biological test cases where we had access to knowledge that allowed us to validate its biological coherence. In this context, our growth rate estimation is the same as in WaddingtonOT¹⁰, with their default parameters on which the authors of WaddingtonOT already performed a sensitivity analysis. In addition, the

same parameters have been used more recently in MOSCOT⁷. The only difference in our growth rate estimation is given by the choice of the deltaT, which is set to the real time distance between time points for WaddingtonOT and it's instead set to 1 for STORIES.

To address the request of the Reviewer, we thus assessed the impact of variations in delta T on the estimated growth rate. delta T has a very simple impact on the estimation as it acts as an inverse temperature parameter. A higher delta T will result in a more entropic distribution of growth rates without changing the ranking of the cells in this distribution. On the other hand, using delta T = 0 leads to estimating a constant growth rate for all cells which is therefore equivalent to not modeling growth.

The growth rate models the potential division (rate > 1) or death (rate < 1) of cells between the measured time points. We relied on prior knowledge of the length of the cell cycle in the biological conditions we studied to make sure that our choice of deltaT was reasonable.

In the axolotl regeneration case, for all tested values of delta T our estimated growth rates were higher for less differentiated cells, which makes biological sense. In addition, we found in the literature that in the context of an injury, axolotl spinal cord cells might take between approximately 5 and 15 days to divide¹¹. This means that cells could divide at most twice between consecutive timepoints in our dataset, thus giving birth to 4 descendants. As shown in Supplementary Figure 18A-C, using delta T = 1 is coherent with this estimation as the maximum estimated growth rate is 4 in this case. While using a higher value for delta T leads to unreasonably high estimated growth rates, using a value between 0 and 1 interpolates between not modeling growth and the upper bound on maximum growth between measured timepoints.

In mouse midbrain development, we also observed that our estimated growth rates were higher for the less differentiated cells: radial glial cells. Moreover, we found in the literature that mouse cells between 13 and 17 embryonic days (in the same range as our dataset which has 3 time points at E12.5, E14.5 and E16.5) take between approximately 16 and 26 hours to divide¹². This means that cells could replicate at most three times between consecutive timepoints in our dataset (which would result in a cell giving birth to 8 descendants). As shown in Supplementary Figure 18D, our estimated growth rates with delta T = 1 are coherent since they are below the theoretical upper bound.

Notably, our estimation of the growth rate is not part of STORIES' package. Indeed, while we allow users to input their growth rate inside the method, we do not restrict the method to any specific strategy to estimate it. To introduce the growth rate in STORIES, we used the most consensual formalization currently available. We believe however that future developments could probably lead to a better modeling of cell death and division processes and for this reason we let the users choose their own modeling strategy.

We now modified the Methods of the paper to better clarify that the growth rate estimation has been only performed in the two test cases and provide the analysis on the impact of delta T on the growth rate estimation in Supplementary Figure 18.

8. Could you clarify computational costs? FGW is computationally costly and I guess incorporating it as a loss term in deep learning further increases the cost. Is the tool

feasible for biologists with normal workstation to analyze a sequence of Stereo-seq data each containing thousands of cells? Adding similar plots as Supplementary Figure 1 but with non-zero alpha will be helpful.

Reply: Since we run on mini-batches, we don't suffer from memory limitations and can handle very large datasets. We highly recommend using STORIES on a device with a GPU since our implementation can leverage GPU acceleration to make the training much more efficient. As a reference, using an Nvidia A40 GPU, running STORIES on the zebrafish training dataset with 8k cells and 5 timepoints takes about 4-11 minutes. The obtained run times are comparable to those observed for PRESCIENT on the same setting (around 10 minutes).

We now added Supplementary Figure 17 showing how computational costs evolve with α . Overall, we can see that while using greater alphas (like 0.1) makes the training longer due to Gromov Wasserstein being a much more computationally heavy problem than its linear counterpart, using a small alpha leads to a more than reasonable training time. Note that, in some datasets, using a low alpha leads to an even faster training than using $\alpha=0$. We could assume that the regularization induced by the quadratic term in FGW makes the problem converge faster as it introduces more constraints on the solution. Furthermore, implementation details of the Optimal Transport library we use (OTT-jax¹³) might impact the runtime of the method.

We now specified the computational cost in the first section of Results and in Methods section "Computational runtime".

9. In evaluation, instead of using Wasserstein distance for gene expression score and GW distance for spatial score, why using FGW with a single alpha parameter to determine both scores? IMHO, the earlier approach is more natural.

Reply: We agree with the reviewer that the Wasserstein distance between predictions and the ground truth, which is the classical evaluation metric for this problem^{1,3-6}, is a more natural choice. We now evaluate performances in the benchmark (Figure 2) using this metric. Regarding the use of GW distance for spatial score, since STORIES does not predict the spatial positions of the next time points but only the gene expression profiles, the GW distance for the spatial score would be exactly the same for all methods. We instead used FGW with a single alpha parameter because it finds a transport plan which solves a tradeoff between gene expression similarity and spatial consistency. From this optimal transport plan, we can then derive gene expression and spatial errors which we now show in Supplementary Figure 2.

We now updated Figure 2 using the Wasserstein distance, the results section 2 and methods section "evaluation" accordingly.

10. While modeling cell state dynamics with Waddington landscape is widely accepted, could you clarify why the spatial component can also be modeled as an energy potential (i.e., curl-free)? It would be helpful to highlight the biological interpretation of this assumption.

Reply: We would like to stress that we do not model the spatial component as an energy potential, the latter is a function of the gene expression only. The space is only used to implicitly enforce spatial consistency the training of the gene expression potential through the spatial term in the FGW loss.

We now further clarified this in the first section of Results: “Of note, this potential is not a function of space but only of gene expression.”

Reviewer #2 (Remarks to the Author):

Summary:

The paper presents SpatioTemporal Omics eneRgIES (STORIES), a novel method designed to infer cell fate trajectories from spatiotemporal transcriptomics data using Fused Gromov-Wasserstein (FGW). This method models cell differentiation by learning a potential landscape that integrates spatial and gene expression data, providing predictions for future cellular states and gene trends. The method is benchmarked on large-scale spatiotemporal datasets, such as axolotl neural regeneration and mouse gliogenesis. The method demonstrates superior spatial coherence relative to its linear non-spatial counterpart and shows it is capable of generating biologically relevant insights.

Strengths:

The method improves upon existing OT methods by integrating both spatial and gene expression data. The method applies an FGW cost which allows the model to incorporate spatial information up to isometries.

Some nice applications are given, including an analysis of trajectories in mouse dorsal midbrain development. A few interesting genes are identified for further investigation in gliogenesis.

The comparison to the linear method clearly highlights the usefulness of incorporating spatial coordinates, showing that STORIES’s inclusion of spatial information enhances spatial coherence and predictive accuracy across multiple datasets. The results emphasize the value of integrating spatial data in the loss function.

The method is provided as an open-source Python package, enabling integration with existing single-cell analysis tools, making it accessible for broader use.

Limitations/Weaknesses/Questions/Comments:

1. As the authors note in the discussion, the potential function does not incorporate spatial data directly and generate predictions in spatial coordinates, so it does not model the physical migration of cells. Also, the use of a potential-based model might oversimplify complex biological processes, such as cell-cell communication or

oscillations within cell states. Although not the primary goal of the article, addressing these aspects would significantly improve the impact of the manuscript.

Reply:

Although we do not directly model the influence of the spatial context of a cell as an input of the gene expression potential, we nonetheless believe that, through the implicit constraint imposed by our FGW loss, our potential benefits from the spatial information. Indeed as shown in Figure 3D, the predictions obtained by STORIES are much more spatially coherent than those of the linear state-of-the-art even though the space is not used at inference time. Furthermore, we showed in the case studies how STORIES' learned potential displays different branching decisions related to the spatial environment. For instance, in the context of axolotl regeneration, STORIES predicted different fates for reEGCs depending on their spatial location in the slice. Importantly, these predictions were coherent with the spatial localizations of differentiated cell populations at later time points. Therefore, despite its potential not explicitly incorporating the spatial information, the aforementioned elements show that STORIES already represents a significant step toward integrating spatial context into trajectory inference.

At the same time, as explained in our Discussion section, we agree with the reviewer that our model's assumptions fail to fully capture complex biological mechanisms such as cell-cell communication or cell state oscillations. Although we aim to address these gaps in future work, doing so is extremely difficult, as we now elaborate on in the Discussion. Achieving this goal requires overcoming two foreseeable yet highly challenging aspects (both numerically and in terms of deriving biological insights):

1. Extending the potential $\phi(x)$ to $\phi(x, s)$: This extension would make the potential depend on both gene expression (x) and spatial position (s). However, this is nontrivial because, unlike gene expression data, the spatial positions of cells at two consecutive time snapshots are not aligned within the same reference frame. Ad-hoc registration methods might not account for the complex deformations that occur in developing organisms as well as Gromov-Wasserstein could. Note however that extending Wasserstein flows to Gromov-Wasserstein flows remains poorly understood theoretically since its introduction by Sturm¹⁴ (see also this recent work¹⁵) and is currently infeasible for practical implementation.

2. Introducing an interaction potential $\phi(x, x')$: This involves replacing the single-cell potential $\phi(x)$ with an interaction potential $\phi(x, x')$. However, this approach presents several challenges. First, the computational complexity of evaluating flows becomes quadratic, which does not scale well for large datasets. Second, and more critically, the parameterization introduces an enormous number of degrees of freedom, making it susceptible to overfitting unless robust regularization techniques are applied. Additionally, the velocity and displacement field for a given cell's location (s) would be influenced by all its neighbors, significantly complicating the process of deriving interpretable biological insights.

For these reasons, while we described these directions in our Discussion section as extremely promising, we believe that they fall outside the scope of this work. At the same time, the aforementioned elements show that STORIES already represents a significant step toward integrating spatial context into trajectory inference. The above discussion on the difficulties related to those extensions is now integrated in our revised manuscript.

2. The method is evaluated on 50 PCA components from 10k highly variable genes. How is training time and performance impacted when using all genes instead of the 10k highly variable ones? How is the training time and performance impacted when using more PCA components than 50?

Reply: Regarding the impact of using 10k highly variable genes vs all genes, for training time there is no impact as the training is performed on PCA-reduced data. So the dimension that the model takes is independent from the number of features prior to the PCA projection. Regarding the impact on the performance, as PCA optimizes for directions with the greatest variance, the least variable genes are not expected to contribute much to the first 50 PCA components. As a consequence, using 10k highly variable genes vs all genes prior to PCA projection is not expected to impact results.

Regarding the choice of the number of PCA components, we realized that our choice of dimensions was not clear. First, the data was reduced to 50PCs, then batch correction was performed with Harmony and finally only the first 20 PCs were kept for training (this last part was mentioned in the method's section "Computational Optimal Transport"). This choice was motivated by pre-existing works using dimensions between 5 and $20^{2-4,16}$.

We then tested the impact of the number of PCA components on both training time and performances. To assess the latter, we used the cell type accuracy metric on the mouse dataset. Indeed, during the development of living organisms some cell types appear, disappear or transform into new cell types. In the mouse dataset the authors of MOSCOT⁷ performed an extensive literature search in order to annotate cell type transitions which were biologically valid. We used this ground truth to evaluate the cell type transitions predicted by STORIES. For this evaluation, we used the gene expression predicted by each method at time $t+1$, then computed a transport plan between this predicted gene expression and the ground truth measurements at $t+1$. This transport plan was then used as a similarity matrix on which we applied k-nn classification. This resulted in cell type predictions which we compared with the ground truth. The use of cell type accuracy was the only option to evaluate the impact of the number of PCA components as our primary evaluation metric, Wasserstein distance between gene expression profiles, is not comparable across different spaces.

As shown in Supplementary Figure 16, using more than 20 PCs does not increase the training time but also increases the performance. As a result of this we now updated all benchmarking results using 50 PCs.

We now updated Figure 2, the results section 2 and the methods section "computational optimal transport" regarding the number of selected PCA components.

3. Y-axis labels for the scatter plots would help readability

Reply: We now updated Figure 3 and 4 to report Y-axis labels in the scatter plots

1. Yeo, G. H. T., Saksena, S. D. & Gifford, D. K. Generative modeling of single-cell time series with PRESCIENT enables prediction of cell trajectories with interventions. *Nat.*

- Commun.* **12**, 3222 (2021).
2. Hashimoto, T., Gifford, D. & Jaakkola, T. Learning Population-Level Diffusions with Generative RNNs. in *Proceedings of The 33rd International Conference on Machine Learning* 2417–2426 (PMLR, 2016).
 3. Bunne, C., Papaxanthos, L., Krause, A. & Cuturi, M. Proximal Optimal Transport Modeling of Population Dynamics. in *Proceedings of The 25th International Conference on Artificial Intelligence and Statistics* 6511–6528 (PMLR, 2022).
 4. Tong, A., Huang, J., Wolf, G., Dijk, D. V. & Krishnaswamy, S. TrajectoryNet: A Dynamic Optimal Transport Network for Modeling Cellular Dynamics. in *Proceedings of the 37th International Conference on Machine Learning* 9526–9536 (PMLR, 2020).
 5. Huguet, G. *et al.* Manifold Interpolating Optimal-Transport Flows for Trajectory Inference. *Adv. Neural Inf. Process. Syst.* **35**, 29705–29718 (2022).
 6. Zhang, J., Larschan, E., Bigness, J. & Singh, R. scNODE: generative model for temporal single cell transcriptomic data prediction. *Bioinformatics* **40**, ii146–ii154 (2024).
 7. Klein, D. *et al.* Mapping cells through time and space with moscot. 2023.05.11.540374 Preprint at <https://doi.org/10.1101/2023.05.11.540374> (2023).
 8. Wei, X. *et al.* Single-cell Stereo-seq reveals induced progenitor cells involved in axolotl brain regeneration. *Science* **377**, eabp9444 (2022).
 9. Weiler, P., Lange, M., Klein, M., Pe'er, D. & Theis, F. CellRank 2: unified fate mapping in multiview single-cell data. *Nat. Methods* **21**, 1196–1205 (2024).
 10. Schiebinger, G. *et al.* Optimal-Transport Analysis of Single-Cell Gene Expression Identifies Developmental Trajectories in Reprogramming. *Cell* **176**, 928-943.e22 (2019).
 11. Cura Costa, E., Otsuki, L., Rodrigo Albors, A., Tanaka, E. M. & Chara, O. Spatiotemporal control of cell cycle acceleration during axolotl spinal cord regeneration. *eLife* **10**, e55665 (2021).
 12. Hoshino, K., Matsuzawa, T. & Murakami, U. Characteristics of the cell cycle of matrix cells in the mouse embryo during histogenesis of telencephalon. *Exp. Cell Res.* **77**, 89–94 (1973).

13. Cuturi, M. *et al.* Optimal Transport Tools (OTT): A JAX Toolbox for all things Wasserstein. Preprint at <https://doi.org/10.48550/arXiv.2201.12324> (2022).
14. Sturm, K.-T. The space of spaces: curvature bounds and gradient flows on the space of metric measure spaces. Preprint at <https://doi.org/10.48550/arXiv.1208.0434> (2020).
15. Zhang, Z., Goldfeld, Z., Greenewald, K., Mroueh, Y. & Sriperumbudur, B. K. Gradient Flows and Riemannian Structure in the Gromov-Wasserstein Geometry. Preprint at <https://doi.org/10.48550/arXiv.2407.11800> (2024).
16. Peng, Q., Zhou, P. & Li, T. stVCR: Reconstructing spatio-temporal dynamics of cell development using optimal transport. 2024.06.02.596937 Preprint at <https://doi.org/10.1101/2024.06.02.596937> (2024).

Reviewer #1 (Remarks to the Author):

The authors have addressed all my concerns. I believe the manuscript has been significantly improved and ready for publication.

Reply : We would like to thank the reviewer for their constructive comments which helped us improve the manuscript.

Reviewer #2 (Remarks to the Author):

Thank you for providing the additional discussion section on potential extensions and their current obstacles, which add color to the contributions of the current manuscript. The authors have clarified my understanding of the role of the FGW loss in the context of incorporating the spatial information in STORIES.

- One additional point of clarification: in the methods section on “Learning the Potential”, is the notation used for the different sets of spatial coordinates, “r” and “s”, meant for a general formulation of the quadratic term in the FGW formulation? And am I correct in that the r and s are the same when optimizing f_{θ} as shown on line 638, since STORIES does not predict explicitly spatial coordinates? Perhaps a sentence just clarifying what r and s are in practice for STORIES would be useful.

The additional metrics provided in response to the other reviewer have enhanced the story in my view. While I agree with the other reviewer that the improvement in the approach is not groundbreaking, it is still a nontrivial improvement, and I believe the method is well-justified and provides a step in the right direction regarding the difficult task of modeling cell fates. I would support acceptance for this manuscript in its current state.

Reply :Regarding the point raised : yes that's exactly right, to introduce the general formulation of FGW we needed to use "r" and "s" to distinguish between the spatial coordinates of the two measures but since we don't predict the spatial coordinates "r" and "s" are the same in our final loss. We modified the text accordingly to clarify this point in the Methods section:

"Our model doesn't predict spatial coordinates therefore r and s are identical in our FGW loss and correspond to the spatial positions of cells at time t_{k-1} ".

Thank you for your insightful comments.